# ACTIVITY-DRIVEN QUANTILE OPTIMIZATION: DYNAMIC EXPLORATION AND EXPLOITATION IN RECOMMENDER SYSTEMS

## ABSTRACT

Recommender systems, as core components of modern digital platforms, leverage reinforcement learning (RL) paradigms to optimize long-term user experience through exploration-exploitation trade-offs. While existing studies implement differentiated policies via coarse-grained user grouping, they face critical challenges in dynamically evolving scenarios: how to capture fine-grained user state transitions and establish precise exploration-exploitation balancing mechanisms. Empirical analysis on existing dataset shows that over $40\%$ of users experience activity level transitions within four weeks, highlighting the need for dynamic optimization. To address this, we propose Activity-Driven Quantile Optimization (ADQO), which integrates a general value critic network for user activity modeling and a quantile critic network to finely characterize the distribution of recommendation values, capturing the stochastic of user feedback. A dynamic policy implements high-potential exploration for low-activity users and low-risk exploitation for high-activity users: optimizing the upper quantiles for low-activity users to uncover latent interests, and the lower quantiles for high-activity users to mitigate risks. We further introduce two alignment losses to enhance training stability and consistency. Experiments demonstrate ADQO's superior performance across three datasets, effectively converting low-activity users to higher states and retaining high-activity users, validating its practical applicability. Our data analysis and training code are shared at https://anonymous.4open.science/r/ADQO-6DC9/.

## 1 INTRODUCTION

Recommender systems (RS) have become a fundamental component of real-world platforms, widely applied in domains such as short-video (Zhan et al., 2022; Zhao et al., 2024; Zhang et al., 2025), e-commerce (Pei et al., 2019; Wang et al., 2024), knowledge sharing (Shen et al., 2009; Zhang et al., 2024b), music streaming (Zhang et al., 2022; Bendada et al., 2020), and news delivery (Wu et al., 2023; Özgöbek et al., 2014). In recent years, reinforcement learning (RL) has been increasingly adopted in RS (Wang et al., 2025; Liu et al., 2023; Zhang et al., 2024a), where its exploration-exploitation trade-off capability continuously balances discovering users' latent interests (exploration) and recommending content aligned with their historical preferences (exploitation) to maximize long-term cumulative rewards from user interactions and strengthen platform engagement. Recent studies have conducted coarse-grained user grouping based on activity levels and applied differentiated policies (Zhang et al., 2024a). However, the evolving user states in large-scale recommendation platforms pose challenges to RL algorithms: continuously capturing dynamic shifts in user states, modeling stochastic user feedback with fine granularity, and establishing effective exploration-exploitation trade-offs, which are essential for real-world applications.

To validate this challenge, we conducted analysis experiments on the widely used short-video dataset KuaiRand[1]. As shown in Figure 1, we divided one month of data into four weeks and analyzed cross-week transitions in user activity levels. Specifically, using the number of clicked items, number of exposed items, and click-through rate (CTR) as activity metrics, users were categorized into

---

[1] https://kuairand.com/

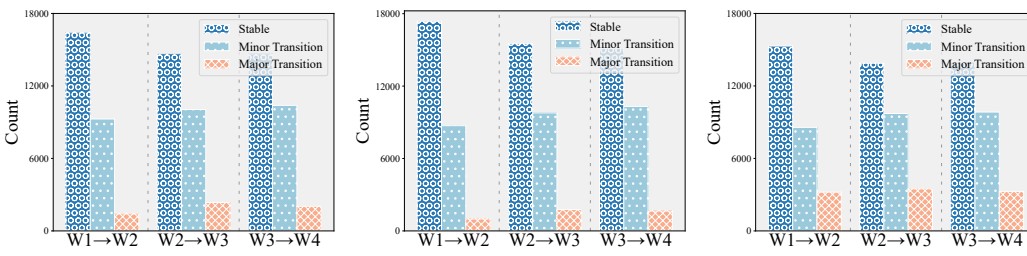

(a) Activity group by click number.    (b) Activity group by exposure.    (c) Activity group by CTR.

Figure 1: Statics of user activity level transitions in 1 month on KuaiRand. We divided one month of data into 4 weeks. Users were classified into three groups (low, medium and high activity) each week based on activity metrics: the number of items clicked, the number of items exposed, or the click-through rate (CTR). For the period from each week $W_i$ to the subsequent week $W_{i+1}$, we analyzed the proportion of users experiencing transitions in activity levels. Mutual transitions between low and medium activity, as well as between medium and high activity, were defined as minor transitions. Mutual transitions between low and high activity were defined as major transitions.

low, medium, and high activity groups. We then quantified the proportion of users experiencing activity transitions from each week to the next, defining three transition types: stable (no change in activity level), minor transitions (mutual shifts between low-medium or medium-high activity), and major transitions (direct shifts between low-high activity). Our observations revealed that while stable users dominated, above $40\%$ of users experienced activity transitions (minor or major). A more detailed analysis is presented in Appendix I. This highlights the critical, unresolved need to dynamically tailor exploration-exploitation trade-offs in a fine-grained manner for diverse users as platform dynamics evolve, which serves as a key frontier for optimizing user experience.

To address this challenge, this work proposes Activity-Driven Quantile Optimization (ADQO). By computing user state values that characterize fine-grained user activity through a general value critic network, it captures the dynamic characteristics of user activity levels in recommendation systems. To simultaneously model the uncertainty of user feedback, an additional quantile critic network is introduced to model the value distribution of items recommended to the current user. Then ADQO implements an adaptive strategy by performing high-potential exploration for low-activity users and low-risk exploitation for high-activity users: the recommendation policy optimizes the upper quantiles of value for low-activity users to uncover latent interests while optimizing the lower quantiles of value for high-activity users to mitigate risks. Furthermore, by integrating an exploration-weighted dual critic alignment mechanism and a guidance-based policy feedback alignment mechanism, the training stability and consistency of the critic networks and policy network are further enhanced. Experiments on three datasets demonstrate that ADQO performs exceptionally well, while analysis shows it can effectively convert more low-activity users into high-activity users and improve the retention rate of high-activity users, highlighting its significant practical application potential.

## 2 PRELIMINARIES

### 2.1 PROBLEM FORMULATION

To optimize the recommendation policy, the problem is modeled as a Markov Decision Process (MDP). In the recommendation system, the user population $U$ serves as the environment, and our policy selects items from the set $I$ to recommend. The MDP is defined by the 5-tuple $\langle S, A, P, R, \pi \rangle$, with each component specified as follows:

- **State Space** $S$: A continuous space where $s_t \in S$ represents the user state, including both static features (e.g., genders) and dynamic features (e.g., interaction history) at each time step $t$.
- **Action Space** $A$: At each time step $t$, the action $a_t \in A$ is a recommendation list of fixed size $K$, formalized as $A = I^K$, where each element in the list is an item from the candidate set $I$.
- **State Transition Function** $P$: Defined as $P : S \times A \times S \to \mathbb{R}$, where $p(s'|s, a)$ denotes the probability of transitioning from the current state $s$ to a new state $s'$ after executing the action $a$.

- **Reward Function** $R$: Defined as $R : S \times A \to \mathbb{R}$, mapping the user state $s$ and recommendation action $a$ to an immediate reward $r(s, a)$, which reflects user feedback (e.g., clicks, purchases).
- **Recommendation Policy** $\pi$: Defined as $\pi : S \to A$, which specifies the action (item list $a$) recommended for user state $s$ at each time step. Given the large action space, we employ a deterministic policy: $\pi(s)$ outputs a relevance vector, computes similarity scores with embeddings of items in $I$ as $P(I_i|\pi(s)) = \text{Emb}_{I_i} \cdot \pi(s)$, and selects the top-$K$ items as the recommended action, as detailed in Appendix D.

The objective is to find a optimal policy $\pi^*$ to maximize the expected cumulative reward:

$$\pi^* = \arg\max_\pi \mathbb{E}\left[\sum_{t=0}^{\infty} \gamma^t r(s_t, a_t)\right], \qquad (1)$$

where the discount factor $\gamma \in [0, 1]$ balances the trade-off between immediate and long-term rewards in the cumulative return calculation.

## 2.2 Distributional Reinforcement Learning

In recommendation systems, expected-value maximization has a critical limitation: User behavior is inherently stochastic, interactions like clicks, purchases, or session durations often fluctuate unpredictably. A user may randomly skip a preferred item or engage with an unexpected one. This stochasticity means scalar expected values alone cannot fully capture the probability distribution of future returns. We therefore need to model the complete distribution of return values. Such distributional modeling reflects not just average returns but also variability, skewness, and extreme cases, enabling more robust decisions that consider both risk and potential. To address this limitation and enable risk- and potential-aware optimization tailored to user activity, we turn to Distributional Reinforcement Learning (DRL), which models the complete distribution of future returns $Z(s, a) = \sum_{t=0}^{\infty} \gamma^t r(s_t, a_t)$, shifting focus from scalar expectations to explicitly characterize reward uncertainty, thereby enabling agents to make distributionally informed decisions Bellemare et al. (2017; 2023). The distributional Bellman equation defines how return distributions evolve:

$$Z(s, a) \overset{D}{=} r(s, a) + \gamma Z(s', a'), \qquad (2)$$

where $s' \sim P(\cdot|s, a)$, $a' \sim \pi(\cdot|a')$, and $\overset{D}{=}$ denotes equality in probability laws. By modeling full return distributions, DRL agents exploit higher-order statistical properties such as variance and skewness, achieving superior performance in environments requiring robustness to uncertainty. Theoretical convergence is guaranteed through probability metrics like the Wasserstein distance.

Building on this foundation, C51 (Bellemare et al., 2017) discretize return distributions into $N$ fixed atoms $\{z_i\}_{i=1}^N$, parameterizing cumulative probabilities $p_i = \mathbb{P}(Z \leq z_i)$. These methods minimize cross-entropy loss between predicted probabilities and projected Bellman targets, with projection ensuring consistency within predefined support boundaries. Quantile regression-based approaches such as QR-DQN (Dabney et al., 2018b) instead directly approximate the quantile function $F_Z^{-1}(\tau)$ ($\tau \in [0, 1]$), which is formally defined as the inverse of the cumulative distribution function (CDF) $F_Z(z) = \mathbb{P}(Z \leq z)$ of the return distribution $Z$. By modeling $F_Z^{-1}(\tau)$, QR-DQN characterize the relationship between quantile levels $\tau$ and their corresponding return thresholds, and optimize the $p$-Wasserstein distance between predicted and target distributions:

$$W_p(U, V) = \left(\int_0^1 \left|F_U^{-1}(\tau) - F_V^{-1}(\tau)\right|^p d\tau\right)^{1/p}.$$

This allows QR-DQN to dynamically adapt quantile positions without fixed support constraints, better approximating complex distributions. Extending QR-DQN, Implicit Quantile Networks (Dabney et al., 2018a) learn implicit representations of full quantile functions rather than discrete quantile sets, enabling flexible approximation of any return distribution. This supports risk-sensitive policies via customizable quantile sampling, boosting adaptability in uncertain environments.

## 3 ADQO: Our Method

We present our ADQO framework in Figure 2. First, we introduce dual critic networks: a value critic to model user value (activity level), and a quantile network optimized via quantile regression

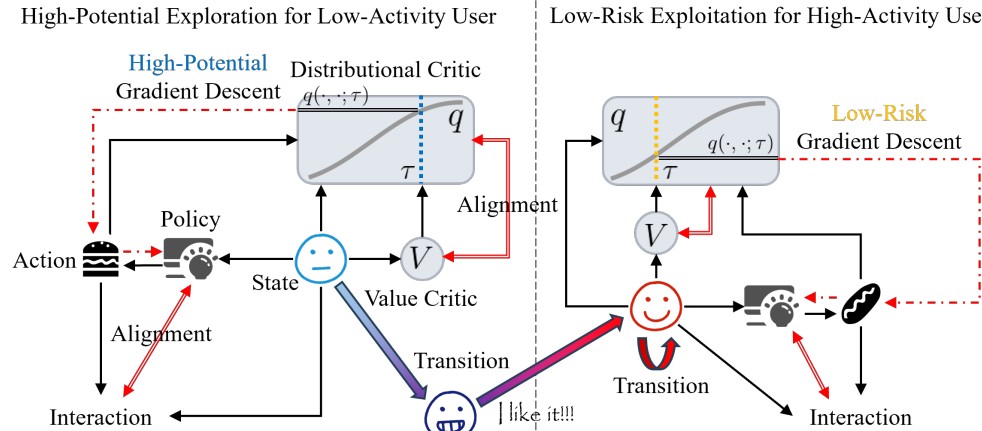

Figure 2: Illustration of ADQO. For each user, activity level is first assessed via a value critic, which is then mapped to the quantile value corresponding to the value distribution modeled by the distributional critic, thereby optimizing the recommendation policy. Additionally, two alignment losses are introduced: one for aligning the value critic with the distributional critic, and another for aligning the policy with user feedback on items. Through this approach, more low-activity users are converted to high-activity status, while preventing high-activity users from churning.

to model the value distribution of recommending an item to the current user under the recommendation policy. For policy optimization, we leverage the activity estimated from the value critic and value distribution estimated from the quantile network to implement high-potential exploration for low-activity users and low-risk exploitation for high-activity users. This dynamic exploration and exploitation strategy facilitates the transition of low-activity users to high-activity states while preventing high-activity users from churning. Finally, we incorporate exploration-weighted dual critic alignment and guidance-based policy feedback alignment to enhance critic training consistency and policy training stability. The detailed implementation of ADQO is provided in Appendix B.

## 3.1 DUAL CRITIC LEARNING FOR ACTIVITY AND DISTRIBUTION MODELING.

In the actor-critic framework algorithm, the critic provides a critical feedback signal for policy optimization by evaluating state and action values (Lillicrap et al., 2015; Schulman et al., 2017). We introduce two critic networks: one is a general value critic network, used to model user activity. The other is a distributional critic network, designed to model the value distribution of the expected return when taking an action in the user state under the current policy. These two networks lay the foundation for subsequent policy optimization. Next, we elaborate on the objectives and training methods of these two critic networks.

**Value Critic for Activity Modeling.** We design a general value critic network to quantify the activity level of user states. Leveraging the temporal difference (TD) learning mechanism. By fusing immediate user feedback (e.g., clicks, purchases) with predicted values of future states, this network captures two core dimensions of user activity: short-term interaction signals and long-term engagement trends. Specifically, the value network defines a state-value function $V_\omega(s)$ parameterized by $\omega$ to estimate $V(s)$, which mathematically represents the expected cumulative discounted return obtainable when the user is in state $s$ and follows the current policy. A higher $V(s)$ indicates stronger potential for sustained interaction in the current state, while a lower value implies the need to adjust the recommendation strategy to re-engage users. Unlike static metrics like CTR, its TD-driven design captures long-term engagement dynamics, making it a robust activity proxy.

The value network $V(s)$ is trained by minimizing the following standard value-based TD loss:

$$\mathcal{L}_{\text{V-TD}} = \mathbb{E}\left[(r(s,a) + \gamma V_\omega(s') - V_\omega(s))^2\right], \tag{3}$$

where $r(s,a)$ denotes the immediate reward from the user's interaction with recommended action $a$ by policy $\pi$ in state $s$, $\gamma$ is the discount factor balancing the weight of immediate rewards against

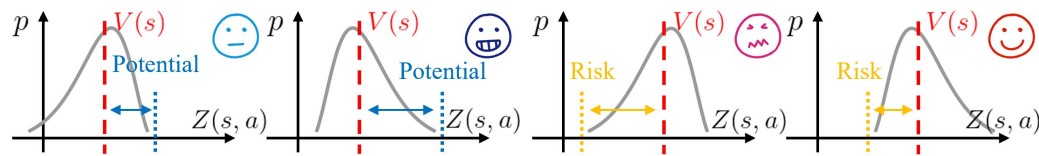

(a) Low activity user with low/high potential action.    (b) High activity user with high/low risk action.

Figure 3: Low-activity users target higher quantiles of $Z(s,a)$ to uncover user potential and conduct effective exploration of engagement. High-activity users avoid lower quantiles of $Z(s,a)$ to mitigate recommendation risks and maintain effective exploitation of engagement.

long-term future returns, and $V_\omega(s')$ is the value estimate of the next state. This loss function operates via the bootstrapping mechanism, using the value prediction of the next state to correct the current-state value estimate, gradually approximating the true expected return.

By optimizing this TD loss, the value network can effectively characterize the mapping relationship between user state $s$ and its estimated value $V_\omega(s)$. This mapping provides a clear quantitative tool for user activity level in policy optimization.

**Quantile Estimation for Distribution Modeling.** As expounded in Section 2.2, we are able to model the value distribution by leveraging quantile regression to learn the quantile function $F_Z^{-1}(\tau)$ ($\tau \in [0,1]$) of the return distribution $Z$. The quantile function is formally defined as the inverse of the cumulative distribution function (CDF) $F_Z(z) = \mathbb{P}(Z \leq z)$, which lays a foundation for subsequent personalized quantile optimization. Specifically, we employ the Implicit Quantile Network $q_\theta(s,a;\tau)$, parameterized by $\theta$, to fit the quantile function $F_{Z(s,a)}^{-1}(\tau)$ of the $Z(s,a)$ distribution. Correspondingly, we derive a modified distributional temporal difference error as follows:

$$\delta_{\tau,\tau'} = r + \gamma q_\theta(s',a';\tau') - q_\theta(s,a;\tau). \tag{4}$$

Next, we utilize the $\tau$-quantile Huber loss for the optimization of $\theta$ defined as:

$$\mathcal{L}_{\text{Q-TD}} = \frac{1}{M^2} \sum_{i=0}^{M} \sum_{j=0}^{M} \rho_{\tau_i}^\kappa(\delta_{\tau_i,\tau_j'}), \tag{5}$$

where $\tau_i$ and $\tau_j'$ sampled independently and uniformly from $[0,1]$ $M$ times, and $\rho_\tau^\kappa(\delta)$ is defined as:

$$\rho_\tau^\kappa(\delta) = \underbrace{\left| \tau - \mathbb{I}_{\{\delta<0\}} \right|}_{\text{quantile loss}} \cdot \underbrace{\begin{cases} \delta^2/2\kappa & \text{if } |\delta| \leq \kappa, \\ |\delta| - \kappa/2 & \text{otherwise,} \end{cases}}_{\text{Huber loss}} \tag{6}$$

where $\kappa > 0$ (usually set to 1) regulates the loss's growth rate. Based on this core component capable of modeling return distributions, we can leverage the estimated quantile values to conduct more fine-grained policy optimization.

## 3.2 ACTIVITY-DRIVEN QUANTILE OPTIMIZATION

By leveraging dual critics to model user activity levels and value distributions respectively, we enable our policy to perform activity-driven quantile optimization. We first provide an intuitive explanation of the policy optimization objective, followed by its technical implementation details.

As illustrated in Figure 3, our distributional critic constructs an estimated quantile network of the return distribution $Z(s,a)$. The optimization strategy follows distinct principles for different users:

- **Low-activity users**: For users with low predicted activity scores shown in Figures 3(a), we prioritize exploration potential. When the action's latent potential is limited, aggressive exploration becomes less effective in boosting activity. Therefore, we optimize for higher quantiles of $Z$ to identify hidden engagement opportunities.

- **High-activity users**: For users with high predicted activity scores shown in Figures 3(b), we emphasize risk-aware exploitation. Actions with high predicted risk may damage user experience through inappropriate recommendations, potentially causing user attrition. Consequently, we avoid lower quantiles of $Z$ to maintain value preservation.

This optimization strategy formalizes the exploration-exploitation trade-off through quantile-specific optimization, where the target quantile selection adapts dynamically based on activity levels.

Next, we derive the calculation of the policy optimization loss function. Although we demonstrate the approach by categorizing users into low- and high-activity groups, we emphasize that no explicit partitioning is performed. For any user state $s$, the value critic first predicts its activity score $V_\omega(s)$, which is then inversely mapped to a quantile $\tau(s) \in [\tau_{\text{low}}, \tau_{\text{high}}]$ through the following mapping:

$$\tau(s) = \tau_{\text{high}} - (\tau_{\text{high}} - \tau_{\text{low}}) \cdot \frac{V_\omega(s) - V_{\min}}{V_{\max} - V_{\min}}, \tag{7}$$

where $0 \leq \tau_{\text{low}} < \tau_{\text{high}} \leq 1$ are predefined bounds. This reverse normalization ensures higher $V_\omega(s)$ values correspond to lower $\tau(s)$ (risk estimates for active users) and lower $V_\omega(s)$ values map to higher $\tau(s)$ (potential estimates for inactive users), achieving smooth interpolation between quantiles based on relative activity levels.

The policy network $\pi_\phi(s)$ parameterized by $\phi$ generates actions $a = \pi_\phi(s)$, which are then evaluated by the quantile critic $q_\theta(s, a; \tau(s))$ using the adaptive quantile level $\tau(s)$ from equation 7. However, directly optimizing this quantile leads to instability in policy network training due to inconsistent training objectives. To address this, we additionally introduce the expectation of the value $\mathbb{E}[Z(s, a)]$, and thus we formulate the policy optimization objective as:

$$\mathcal{L}_{\text{Policy}} = - \left( q_\theta(s, a; \tau(s)) + \frac{1}{N} \sum_{n=1}^{N} q_\theta(s, a; \tau_n) \right), \quad \tau_n \sim \mathbb{U}(0, 1), \tag{8}$$

where $N$ is the sample number and $\tau_n$ is uniformly sampled from $[0, 1]$ to approximate expectation.

By optimizing the policy network parameters $\phi$ through this loss function, we obtain the desired policy that performs high-potential exploration for users with lower activity levels and low-risk exploitation for users with higher activity levels.

### 3.3 Alignment of Dual Critic and User Feedback

After establishing the dual critics and dynamic quantile policy optimization framework, we introduce exploration-weighted dual critic alignment to enhance critic training consistency and guidance-based policy feedback alignment for policy training stability.

**Exploration-Weighted Dual Critic Alignment.** For a given state $s$, the previously defined $V(s)$ (maximum expected return across all actions) and $Z(s, a = \pi(s))$ (expected return of the quantile-optimal action) may misalign. This discrepancy stems from the policy's dynamic exploration-exploitation mechanism, which introduces inherent stochasticity into the critic networks and disrupts consistency between these two value metrics. To address this, we design an exploration-weighted dual critic alignment loss. This loss aligns the expected value of $Z(s, a)$ with $V(s)$ while imposing structural constraints, effectively reconciling the stochasticity-induced misalignment and preserving the integrity of value estimation.

First, the exploratory nature of an action can be measured by the difference between the predefined $\tau_{high}$ and $\tau_{low}$:

$$\Delta Z(s, a) = q_\theta(s, a; \tau_{high}) - q_\theta(s, a; \tau_{low}), \quad \tau_{high} > \tau_{low}. \tag{9}$$

A larger $\Delta Z(s, a)$ indicates a wider $Z$-value distribution for the policy's exploratory actions, meaning stronger exploratory nature.

For a batch of $B$ state-action pairs, the dual critic alignment loss is defined as:

$$\mathcal{L}_{\text{C-align}} = \frac{1}{B^2} \sum_{i,j=1}^{B} \max \left( 0, \epsilon - (\mathbb{E}[Z(s_i, a_i)] - \mathbb{E}[Z(s_j, a_j)]) \cdot (V(s_i) - V(s_j)) \right) \cdot w_{ij}, \tag{10}$$

where $\epsilon = 0.1$ controls the margin width, and the expectation $\mathbb{E}[Z(s, a)]$ is also approximated via sampling in the same manner as in equation 8. The adaptive weight $w_{ij}$ combines exploration intensities from both samples with stop-gradient:

$$w_{ij} = 1 - \max (\Delta Z(s_i, a_i), \Delta Z(s_j, a_j))_{\text{stop-gradient}}. \tag{11}$$

This design adaptively relaxes alignment constraints for state-action pairs with strong exploratory characteristics while preserving precise ordinal matching for exploitative decisions.

**Policy Alignment with User Feedback.** Additionally, to effectively guide the direction of policy exploration and credit assignment, we introduce a supervised learning-based feedback alignment module $\mathcal{L}_{\text{Align-F}}$ to align the items recommended by the policy at each step with user feedback. Although this approach is suboptimal due to its neglect of long-term value modeling, it still contributes to the stability of policy optimization. We define it using a cross-entropy loss as follows:

$$\mathcal{L}_{\text{F-align}} = -\frac{1}{T}\sum_{t=1}^{T}\left[-y_t \log P\left(I_t \mid \pi_\phi(s_t)\right) - (1-y_t)\log\left(1 - P(I_t \mid \pi_\phi(s_t))\right)\right], \quad (12)$$

where $y_t \in \{0,1\}$ is the binary user feedback at time step $t$, $P\left(I_t \mid \pi_\phi(s_t)\right)$ is the probability of selecting exposure item $I_t$ given the deterministic action $\pi_\phi(s_t)$, as mentioned in Section 2.1.

## 4 EXPERIMENTS

### 4.1 EXPERIMENTAL SETTINGS

**Datasets.** We include three public datasets: The **KuaiRand-1K**[2] dataset contains multi-behavior interaction logs sampled from 1,000 users for short videos on Kuaishou, a video-sharing mobile app. **ML-1M**[3], a subset of the MovieLens dataset, includes 1 million movie ratings. **RL4RS**[4] is the first open-source real-world dataset specifically designed for reinforcement learning-based recommendation systems. All datasets are split into training and test sets based on timestamps, with the first 75% allocated for training and the remaining 25% for testing. Detailed preprocessing steps and statistical summaries are provided in Appendix C.1.

**Online Simulator and Evaluation Metrics.** To address the limitations of traditional offline metrics in reflecting user feedback under unknown distributions, we introduce an online simulator built on a pre-trained user response model to generate real-time environmental feedback for recommendation policies. Following prior works (Wang et al., 2025; Zhang et al., 2024a; Liu et al., 2023), the simulator assigns a reward of $+1.0$ to items with positive feedback and imposes a penalty of $-0.2$ for negative feedback. Each user session starts with an identical initial temper, which decays incrementally based on recommendation quality until reaching a termination threshold or the maximum interaction limit. Key evaluation metrics include the average total reward per user session, defined as **Total Reward**, and interaction **Depth**, measured by session length. Higher values for both metrics indicate superior long-term recommendation performance and user retention. Full simulator configuration and session designs are documented in Appendix C.2.

**Baselines.** We compare ADQO with several baseline methods: a supervised learning model based on SASRec (Kang & McAuley, 2018), classical reinforcement learning algorithms including A2C (Mnih et al., 2016), DDPG (Lillicrap et al., 2015), TD3 (Fujimoto et al., 2018), and IQN (Dabney et al., 2018a), and RL methods tailored for recommendation systems such as VUERS for user exploration to improve serendipity and long-term user experience (Chen et al., 2021), Wolpertinger (abbreviated as WPG) for large discrete action spaces (Dulac-Arnold et al., 2015), HAC for hyperparameter-space exploration (Liu et al., 2023), DEHAC integrating user and policy stochasticity (Wang et al., 2025), and UOEP leveraging user activity-level grouping for exploration (Zhang et al., 2024a). Additional implementation details are provided in Appendix C.3.

**Implementation Details.** For ADQO, we adopt DDPG (Lillicrap et al., 2015) as the backbone framework. Target networks are established for both the critic networks and the policy network, with soft updates implemented to synchronize parameters (detailed implementation is provided in Appendix B). All methods utilize an experience replay buffer to support off-policy training. To ensure convergence, all models are trained for 50,000 epochs. The architectures of the critic and policy networks are kept consistent across methods. Hyperparameters and optimizer configurations are determined via grid search, and the optimal hyperparameter combinations are selected for reporting results. Specific details of model architectures and training configuration are provided in Appendix D and Appendix E.

---

[2]https://kuairand.com/

[3]https://grouplens.org/datasets/movielens/1m/

[4]https://github.com/fuxiAIlab/RL4RS

Table 1: Comparative analysis of ADQO against baselines across three datasets. Superior performance is denoted by bold text, with secondary underlined. * indicates statistically significant improvements (paired *t*-test, $\alpha = 0.05$). Experiments are repeated 5 times with different random seeds, and results report the mean $\pm$ standard deviation.

| Method | KuaiRand | | ML-1M | | RL4RS | |
|---|---|---|---|---|---|---|
| | Total Reward | Depth | Total Reward | Depth | Total Reward | Depth |
| SL | $13.31 \pm 0.19$ | $14.12 \pm 0.17$ | $14.21 \pm 0.73$ | $14.93 \pm 0.62$ | $8.40 \pm 0.43$ | $9.72 \pm 0.39$ |
| A2C | $10.35 \pm 1.15$ | $11.21 \pm 1.08$ | $14.71 \pm 0.67$ | $15.25 \pm 0.63$ | $7.55 \pm 0.29$ | $8.88 \pm 0.28$ |
| DDPG | $10.06 \pm 1.05$ | $11.02 \pm 0.96$ | $15.75 \pm 1.22$ | $16.30 \pm 1.19$ | $8.05 \pm 1.33$ | $9.38 \pm 1.24$ |
| TD3 | $11.47 \pm 1.68$ | $12.36 \pm 1.53$ | $15.20 \pm 0.85$ | $16.01 \pm 0.80$ | $7.92 \pm 0.71$ | $9.25 \pm 0.65$ |
| IQN | $10.41 \pm 0.25$ | $11.45 \pm 0.24$ | $15.21 \pm 1.12$ | $16.05 \pm 1.05$ | $8.36 \pm 0.62$ | $8.51 \pm 0.59$ |
| VUERS | $11.54 \pm 0.93$ | $12.50 \pm 0.79$ | $14.97 \pm 1.01$ | $15.51 \pm 0.98$ | $8.50 \pm 1.49$ | $9.86 \pm 1.32$ |
| WPG | $12.25 \pm 1.12$ | $13.20 \pm 1.08$ | $15.57 \pm 0.92$ | $16.14 \pm 0.85$ | $8.70 \pm 0.58$ | $10.05 \pm 0.55$ |
| HAC | $12.80 \pm 0.75$ | $13.74 \pm 0.71$ | $15.66 \pm 0.78$ | $16.40 \pm 0.69$ | $8.30 \pm 0.91$ | $9.66 \pm 0.81$ |
| DEHAC | $13.28 \pm 0.88$ | $14.13 \pm 0.84$ | $15.92 \pm 0.72$ | $16.51 \pm 0.64$ | $8.50 \pm 1.08$ | $9.85 \pm 1.02$ |
| UOEP | $\underline{14.32} \pm 0.33$ | $\underline{14.95} \pm 0.31$ | $\underline{16.16} \pm 0.31$ | $\underline{16.78} \pm 0.27$ | $\underline{8.91} \pm 0.82$ | $\underline{10.22} \pm 0.78$ |
| ADQO | $\mathbf{14.94^* \pm 0.31}$ | $\mathbf{15.51^* \pm 0.28}$ | $\mathbf{16.65^* \pm 0.21}$ | $\mathbf{17.28^* \pm 0.20}$ | $\mathbf{9.31^* \pm 0.62}$ | $\mathbf{10.62^* \pm 0.58}$ |

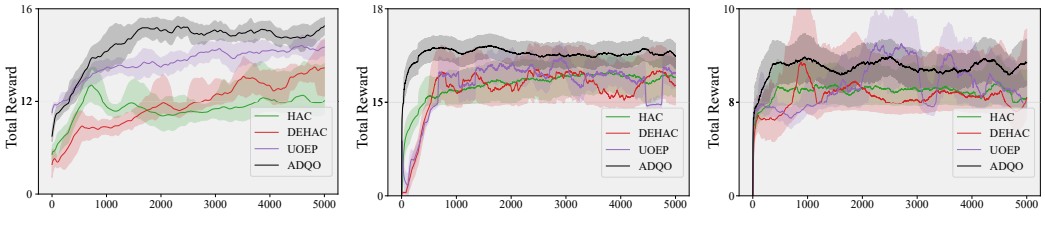

(a) Learing curves on KuaiRand.  (b) Learing curves on Ml-1M.  (c) Learing curves on RL4RS.

Figure 4: Learning curves for HAC, DEHAC, UOEP and ADQO.

## 4.2 MAIN RESULTS

We report the performance results in Table 1. The main findings are as follows: Classical reinforcement learning algorithms perform well, but on large action space datasets such as KuaiRand, some algorithms perform slightly worse than supervised learning. WPG, which is designed for large action spaces, and HAC, which explores hyper-action spaces, achieve better performance due to their exploration properties. VUERS only connects with users via diversity calculations, and blindly boosting diversity harms real-world performance, unlike data-driven exploration based on long-term performance. DEHAC addresses randomness in complex environments by decomposing TD loss. UOEP performs suboptimally because it partitions users into groups based on activity levels and assigns independent policies, leading to high computational and storage costs and reducing scalability. Our ADQO assigns different quantile optimization targets to users with varying activity levels through a single policy network, improving performance while reducing overhead. We also provide the learning curves of ADQO and several baselines in Figure 4, where UOEP plots the best actor. It can be observed that ADQO converges faster and more stably compared with the other three baseline methods. This further highlights the effectiveness of our quantile optimization approach.

## 4.3 ABLATION STUDIES AND FURTHER ANALYSIS

**User Group Validation.** For further validation, we separately selected four user groups following Figure 1 in KuaiRand: stable inactive users (Stable Low), inactive users transitioning to higher activity (Potential

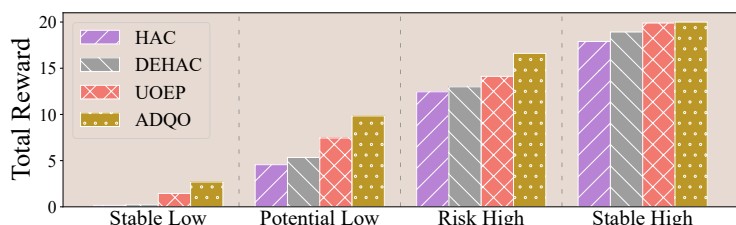

Figure 5: Validation on different user groups.

Low), high-activity users transitioning to lower activity (Risk High), and stable high-activity users (Stable High). Testing the performance of these groups (as shown in Figure 5) reveals that ADQO significantly enhances the activity of low-activity users with upward potential and effectively retains high-activity users at risk of churn, confirming the effectiveness of ADQO.

**Visualization of Exploration and Exploitation.** We used t-SNE embeddings (Van der Maaten & Hinton, 2008) to visualize the action embeddings of a subset of low- and high-activity users in Figure 6. The results show that for low-activity users, the model explores a broader range of action spaces, covering more latent interest areas. In contrast, for high-activity users, the model tends to generate more conservative recommendations within a constrained space, minimizing risks while maintaining user activity.

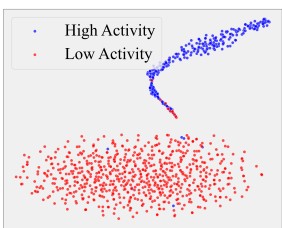

Figure 6: TSNE of actions.

**Ablation of Alignment losses.** ADQO includes only two alignable loss components that can be ablated. We individually disabled $\mathcal{L}_{\text{C-align}}$, $\mathcal{L}_{\text{F-align}}$, and both losses simultaneously. As shown in Table 2, removing either single alignment loss leads to notable performance degradation, with the combined removal resulting in a more pronounced negative impact. This finding indicates that the dual critic networks need to maintain consistency, and the introduction of feedback contributes to enhancing the stability of policy training.

Table 2: Ablations for losses on KuaiRand.

|  | Total Reward | Depth |
|---|---|---|
| ADQO | $\mathbf{14.94 \pm 0.31}$ | $\mathbf{15.51 \pm 0.28}$ |
| w/o $\mathcal{L}_{\text{C-align}}$ | $14.33 \pm 0.23$ | $14.99 \pm 0.19$ |
| w/o $\mathcal{L}_{\text{F-align}}$ | $13.61 \pm 1.22$ | $14.28 \pm 1.06$ |
| w/o both $\mathcal{L}_{\text{align}}$ | $12.38 \pm 1.02$ | $13.41 \pm 0.91$ |

**Sensitivity Analysis of Hyperparameters.** We present the sensitivity analysis in Appendix F.

## 5 RELATED WORKS

*Reinforcement learning* (RL) models *recommendation* as a Markov Decision Process (MDP) combined with deep neural networks into DRL frameworks, establishing core adaptive recommendation technology. Early work framed recommendations as sequential state-action-reward decisions (Shani et al., 2005). Later work includes value-based methods (e.g., DQN for news by Zheng et al. (2018)), policy-based approaches (e.g., actor-critic in Liu et al. (2018)), and techniques for large action spaces (Dulac-Arnold et al., 2015). RL achieves industrial scalability: modified REINFORCE enabled batch RL for billion-user systems (Chen et al., 2019), DDPG-FBE combined policy gradients with ranking for list optimization (Hu et al., 2018), and CMR introduced training-free multi-objective RL for reranking (Chen et al., 2023). DEHAC disentangles TD loss to model policy-user uncertainties (Wang et al., 2025). Specialized methods like RLUR (Cai et al., 2023) and OSPIA (Xu et al., 2023) tackle delayed rewards and instability in user retention.

For the *exploration-exploitation trade-off*, classical approaches include $\epsilon$-greedy (Sutton et al., 1998), UCB (Auer et al., 2002), and parameter noise injection (Plappert et al., 2017). Recent advances feature hierarchical action filtering (Steccanella et al., 2020), information-theoretic rewards (Houthooft et al., 2016), and constrained actor-critic frameworks for multi-objective alignment (Achiam et al., 2017). For recommendation challenges like large discrete action spaces and dynamic preferences, Wolpertinger improves high-dimensional exploration via action embedding and approximate nearest-neighbor search (Dulac-Arnold et al., 2015), while HAC employs two-stage hyper-action inference and kernel-mapped action selection for policy consistency (Liu et al., 2023). UOEP balances interest discovery and stability with activity-level group strategies (Zhang et al., 2024a). However, critical challenges persist in real-world RL-based recommendation systems: continuous transitions in user states demand fine-grained state tracking capabilities, while high-dimensional discrete action spaces and sparse reward signals exacerbate the difficulty of exploration-exploitation trade-offs, calling for more effective algorithmic frameworks to achieve dynamic and stable long-term policy optimization.

## 6 CONCLUSIONS

Recommender systems using RL face challenges in dynamically balancing exploration-exploitation for evolving user activities. Our work addresses this by proposing ADQO, which models fine-grained user activity transitions by dual critic networks and a dynamic policy: exploring high-potential interests for low-activity users and reducing risks for high-activity ones. Experiments show ADQO outperforms baselines, effectively converting low-activity users and retaining high-activity ones, validating its practical value in handling dynamic user behavior.

## REPRODUCIBILITY STATEMENT

We have taken comprehensive measures to ensure the reproducibility of our results. Our data analysis and training code are shared at `https://anonymous.4open.science/r/ADQO-6DC9/`, which includes all necessary scripts, dependencies, and step-by-step instructions for replicating our experiments.

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

## A LLMS USAGE DISCLOSURE

Throughout the writing process, we employed LLMs as an AI-powered writing assistant exclusively to enhance grammatical correctness and textual readability. The tool aided in refining sentence structures and polishing wording, yet it did not contribute to generating technical content, conducting data analysis, or designing algorithms.

## B ALGORITHM IMPLEMENTATION OF ADQO

For the practical implementation of ADQO, we employ an experience replay buffer for off-policy learning and adopt standard DDPG conventions by maintaining target networks for all critic and policy networks with soft update mechanisms (Lillicrap et al., 2015). We present the complete pseudo-code in Algorithm 1.

---

**Algorithm 1** Activity-Driven Quantile Optimization

---

**Require:** Value critic network $V_\omega(s)$ parameterized by $\omega$, distributional critic network $q_\theta(s, a; \tau)$ parameterized by $\theta$, policy network $\pi_\phi(s)$ parameterized by $\phi$, replay buffer $\mathcal{B}$, learning rate $\eta_1$ for critic networks, $\eta_2$ for policy network, $\eta_C$ for $\mathcal{L}_{\text{C-align}}$, $\eta_F$ for $\mathcal{L}_{\text{F-align}}$, soft update parameter $\mu$, target networks $V_{\omega'}(s)$, $q_{\theta'}(s, a; \tau)$ and $\pi_{\phi'}(s)$.
1: Initialize target networks $\omega' \leftarrow \omega$, $\theta' \leftarrow \theta$, $\phi' \leftarrow \phi$.
2: Initialize replay buffer $\mathcal{B}$.
3: **while** not converged **do**
4:     Selection action $a = \pi_\phi(s)$ with current user state $s$.
5:     Execute action $a$ in the environment and observe reward $r$ and new state $s'$.
6:     Store transition $(s, a, r, s')$ in $\mathcal{B}$.
7:     Randomly sample a batch of transitions $(s_i, a_i, r_i, s_i')$ from $\mathcal{B}$.
8:     Compute value critic loss and distributional critic loss $\mathcal{L}_{\text{V-TD}}$ and $\mathcal{L}_{\text{Q-TD}}$ in equation 3 and equation 5.
9:     Compute alignment losses $\mathcal{L}_{\text{C-align}}$ and $\mathcal{L}_{\text{F-align}}$ in equation 10 and equation 12.
10:     Get the estimated activity level $V_\omega(s_i)$ with value critic network.
11:     Calculate $\tau(s_i)$ in equation 7 with $V_\omega(s_i)$ .
12:     Compute $\mathcal{L}_{\text{Policy}}$ in equation 8 with $s_i$, $a_i$ and $\tau(s_i)$.
13:     Gradient step critic networks $\omega, \theta \leftarrow \omega, \theta - \eta_1 \nabla(\mathcal{L}_{\text{V-TD}} + \mathcal{L}_{\text{Q-TD}}) - \eta_C \nabla \mathcal{L}_{\text{C-align}}$.
14:     Gradient step policy network $\phi \leftarrow \phi - \eta_2 \nabla \mathcal{L}_{\text{Policy}} - \eta_F \nabla \mathcal{L}_{\text{F-align}}$.
15:     Soft update target networks $\omega' \leftarrow \mu\omega + (1-\mu)\omega'$, $\theta' \leftarrow \mu\theta + (1-\mu)\theta'$, $\phi' \leftarrow \mu\phi + (1-\mu)\phi'$.
16: **end while**

---

## C EXPERIMENTAL DETAILS

### C.1 PREPROCESSING PROCESS AND STATISTICS OF DATASETS

Table 3: Statistics of datasets.

|                | KuaiRand | MovieLens-1M | RL4RS   |
| -------------- | -------- | ------------ | ------- |
| # Users        | 986      | 6041         | -       |
| # Items        | 11,643   | 3,953        | 283     |
| # Interactions | 96,532   | 97,382       | 781,367 |
| List size      | 10       | 10           | 9       |

Following a standardized preprocessing pipeline, the three datasets are converted into a unified format, with records organized chronologically to incorporate user features, historical interactions, exposed items, user feedback, and timestamps. Data splitting is performed based on timestamps, allocating the first 75% of records for training and the remaining 25% for evaluation.

Following prior work (Liu et al., 2023; Zhang et al., 2024a), positive and negative samples are defined as follows: For the MovieLens-1M dataset, user ratings of 4 or 5 are treated as positive

feedback, while all other ratings constitute negative samples. For the KuaiRand dataset, videos with fewer than 50 interactions are first filtered out, remaining videos with a watch time exceeding 80% of their duration are classified as positive samples, with the rest labeled as negative. Additionally, each user's interaction history in both datasets is segmented chronologically into item lists of length 10, with only positive interactions preceding each segmented list retained as historical behavior to inform the model. The resulting formatted data adheres to the same structural specifications as the RL4RS dataset, ensuring consistency across experimental setups.

Table 3 summarizes the statistics of the preprocessed datasets. Notably, the RL4RS dataset provides user profile features instead of explicit user identifiers, which means a unique user count is not applicable for this dataset.

Table 4: AUC of the simulator on three datasets.

|     | KuaiRand | MovieLens-1M | RL4RS |
| --- | --- | --- | --- |
| AUC | $0.6873 \pm 0.0013$ | $0.6952 \pm 0.0021$ | $0.7674 \pm 0.0011$ |

### C.2 ONLINE SIMULATOR AND SESSION MECHANISMS

To model dynamic user interactions, we develop online simulators from the datasets, designing them to capture reward signals generated in each recommendation round. For each dataset, a user response model $\Psi : \mathcal{S} \times \mathcal{A} \to \mathbb{R}^n$ is trained to estimate the likelihood of user engagement, where the user state $\mathcal{S}$ integrates static attributes (e.g., demographics) and dynamic interaction histories. This model outputs per-item probabilities of positive feedback for the recommended list $a_t$, from which the binary response vector $\mathbf{y}_t \in \{0, 1\}^n$ (e.g., click/no-click) is stochastically sampled to simulate real-world user behavior. We report the AUC of the simulator on three datasets, as shown in Figure 4.

The reward function $r(s_t, a_t)$ is defined as the average feedback across all items in the recommended list: clicked items contribute a reward of 1, while unclicked items yield $-0.2$. This design balances immediate gains and penalties to reflect user satisfaction.

To model session termination, each user begins with an initial "temper" value that decays incrementally based on recommendation quality. Poor recommendations accelerate temper depletion, once the temper drops to zero or below, the session ends. Additionally, sessions are constrained by a maximum depth of 20 interactions, ensuring alignment with typical user engagement patterns and preventing unbounded sequences.

### C.3 DETAILS OF BASELINES

We compare ADQO with several baseline methods:

- **Supervised Learning**: The model is build based on SASRec (Kang & McAuley, 2018), built upon the SASRec architecture, is optimized using observed exposure (recommended item lists) and corresponding user feedback from the training set. Offline training employs a binary cross-entropy loss to align predicted interactions with historical data.

- **A2C** (Mnih et al., 2016): An actor-critic reinforcement learning algorithm that integrates policy gradient and value-based approaches. It simultaneously learns a stochastic policy (actor) and a state-value function (critic) through asynchronous updates, enabling efficient exploration in discrete action spaces.

- **DDPG** (Lillicrap et al., 2015): An off-policy actor-critic method designed for continuous action spaces. Using deterministic policy gradients, it learns a stable policy mapping from states to actions alongside a critic that estimates the expected return, suitable for high-dimensional control tasks.

- **TD3** (Fujimoto et al., 2018): An advanced variant of DDPG that mitigates value function overestimation through twin critics (two value networks) and delayed policy updates. This design enhances stability and performance in continuous action environments, particularly reducing variance in reward estimation.

- **IQN** (Dabney et al., 2018a): A distributional reinforcement learning framework that applies quantile regression to model the full return distribution of state-action pairs. By reparameterizing sample spaces, it implicitly defines a flexible return distribution, enabling the derivation of risk-sensitive policies.

- **VUERS** Chen et al. (2021): An RL method for recommendation systems that employs two core exploration strategies: entropy regularization, a common reinforcement learning trick already included in our baselines like A2C, and intrinsic motivation, incorporating diversity or novelty into rewards. We implement the latter, adding category-based diversity rewards.

- **Wolpertinger** (Dulac-Arnold et al., 2015): A scalable approach for large discrete action spaces, embedding actions into a continuous latent space and using approximate nearest-neighbor search for efficient action selection. This allows effective exploration-exploitation balance in environments with thousands of discrete choices, such as recommendation systems.

- **HAC** (Liu et al., 2023): A hierarchical actor-critic framework decomposing list recommendation into two stages: hyper-action inference (high-level strategy) and effect-action selection (low-level item choice). It incorporates an alignment module and kernel mapping to ensure consistency between abstract strategies and concrete actions, improving policy generalization.

- **DEHAC** (Wang et al., 2025): A disentangled exploration framework for sequential recommendation, addressing the "Mixing Random Factors" challenge in temporal difference learning. By separating the stochastic policy and user environment dynamics in the TD loss, it achieves more accurate long-term reward estimation and robust action exploration, leading to faster convergence and improved policy stability.

- **UOEP** (Zhang et al., 2024a): A user-centric exploration policy designed for heterogeneous user groups. Leveraging a distributional critic to model cumulative reward feedback at varying quantile levels (representing different activity tiers), it deploys a population of specialized actors to perform fine-grained exploration within each user segment. This adaptive strategy enhances long-term user engagement by tailoring exploration intensity to individual activity patterns.

## D MODEL ARCHITECTURES

**Actor Architecture** All RL-based approaches employ SASRec (Kang & McAuley, 2018) as the foundational actor framework. For a given user's historical interaction sequence, the item encoder maps each item to a 32-dimensional embedding, augmented with learnable positional embeddings to capture sequential order. This encoded sequence is fed into a 2-layer Transformer encoder with 4 attention heads and a 0.1 dropout rate, producing a contextual vector representing the user state. This vector undergoes further refinement through two hidden layers (256 and 64 units) to produce the final action vector representation. The final recommendation list is generated by computing dot products between this action vector and all item embeddings, selecting the top-$k$ items with the highest scores.

**Deterministic Critic** The critic for deterministic policies (e.g., DDPG, TD3) utilizes a multi-layer perceptron (MLP) that takes as inputs the encoded user state vector and the action vector derived from the actor. Comprising two hidden layers of dimensions 256 and 64, this network outputs a scalar Q-value representing the expected return of the given state-action pair.

**Value Critic** The A2C value critic follows a similar MLP architecture but differs in input: it receives only the encoded user state vector (excluding the action vector). With identical hidden layer configurations (256 and 64 units), this network estimates the state-value function $V(s)$, which quantifies the expected cumulative reward starting from the current state.

**Distributional Critic** Serving as the critic for IQN, UOEP, and ADQO, this network processes three inputs: the encoded user state, action vector, and quantile parameters. The state and action vectors first pass through two shared hidden layers (256 and 64 units), yielding a 16-dimensional intermediate vector. Concurrently, the quantile values are transformed via a single MLP layer into a 16-dimensional embedding, which is element-wise multiplied with the intermediate vector. The resulting representation is further refined through a 32-unit hidden layer to output the estimated quantile values of the return distribution, enabling quantile optimization.

## E  TRAINING CONFIGURATION

For all methods and experiments, we use a single GeForce RTX 3090 GPU environment for training and perform a grid search over hyperparameters to determine the optimal configuration, with the best-performing settings selected for evaluation. Specifically, the training configuration for ADQO on the KuaiRand dataset is listed in Table 5. All models were developed using the PyTorch framework to ensure reproducibility and compatibility across different experimental setups.

Table 5: Training configuration of ADQO.

| Key | Value |
|---|---|
| GPU | GeForce RTX 3090 (24.00GiB) |
| CPU | Intel(R) Xeon(R) Gold 5218 CPU @ 2.30GHz |
| Operator System | Ubuntu 18.04.5 LTS |
| Memory usage | 4,378MiB /24.00GiB |
| Time of Execution | $\sim 210$ minites |
| Optimizer | Adam (Kingma & Ba, 2014) |
| Batch size | 1,024 |
| Gradient clipping | False |
| Number of epoch | 50,000 |
| $\gamma$: Discount factor | 0.9 |
| $\eta_1$: Critic learning rate | $\{1e^{-4}, 5e^{-4}, \mathbf{1e^{-3}}\}$ |
| $\eta_2$: Policy learning rate | $\{1e^{-4}, \mathbf{5e^{-4}}, 1e^{-3}\}$ |
| $\eta_C$: Critic alignment loss learning rate | $\{1e^{-4}, \mathbf{5e^{-4}}, 1e^{-3}\}$ |
| $\eta_F$: Policy alignment loss learning rate | $\{\mathbf{1e^{-4}}, 5e^{-4}, 1e^{-3}\}$ |
| $\mu$: Target update rate | $1e^{-2}$ |
| $\tau_{\text{low}}$: Lower quantile bound | $\{0.2, 0.25, \mathbf{0.3}, 0.35, 0.4\}$ |
| $\tau_{\text{high}}$: Upper quantile bound | $\{0.8, 0.75, \mathbf{0.7}, 0.65, 0.6\}$ |
| $M$: Sample number of $\tau$ for $\mathcal{L}_{\text{Q-TD}}$ | 32 |
| $N$: Sample number of $\tau$ for $\mathcal{L}_{\text{Policy}}$ | 16 |
| $\lambda$: Regularization loss coefficient | 32 |

## F  SENSITIVITY ANALYSIS OF HYPERPARAMETERS

**Impact of $\tau$.** For $\tau_{\text{low}}$ (lower threshold) and $\tau_{\text{high}}$ (upper threshold), we set $\tau_{\text{high}} = 1 - \tau_{\text{low}}$. A smaller $\tau_{\text{low}}$ reduces the risk of policy loss in high-activity optimization and increases the potential for mining low-activity optimization (as illustrated in Figure 7). Nevertheless, when $\tau_{\text{low}}$ is further reduced, the performance improvement of the model becomes limited, which may stem from inconsistencies in the optimization objectives of the policy network.

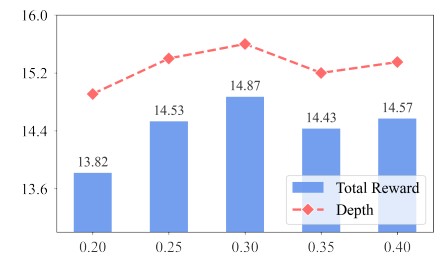

Figure 7: Impact of $\tau_{\text{low}}$.

**Impact of $\eta$ of alignment losses.** We additionally analyzed the learning rates $\eta_C$ and $\eta_F$ assigned to the two alignment losses $\mathcal{L}_{\text{C-align}}$ and $\mathcal{L}_{\text{F-align}}$. As shown in Figure 8, for $\eta_C$, we found that increasing it ensures the consistency of the critic network, thereby improving performance. However, excessive increase leads to performance degradation, possibly due to in-

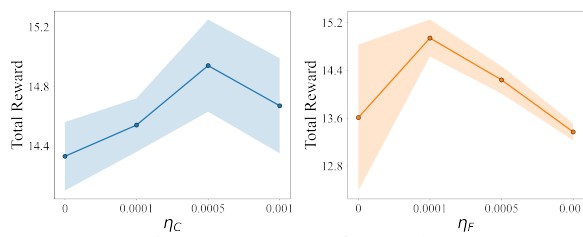

Figure 8: Impact of $\eta_C$ and $\eta_F$.

consistencies caused by over-constraining action exploration. For $\eta_F$, we observed a more significant impact on performance, as it effectively guides the exploration direction in the action space—a phenomenon verified by the reduction in variance as $\eta_F$ increases. However, excessively increasing $\eta_F$ will restrict exploration and degrade performance.

## G    RELATIONSHIP BETWEEN QUANTILES AND USER ACTIVITY

During training, we tracked the quantiles $\tau$ corresponding to users with different activity levels and analyzed their correlation with the value function $V(s)$. We partitioned $V(s)$ into intervals and computed the mean $\tau$ for each interval.

Table 6: Mean of quantile $\tau$ for different $V(s)$ ranges.

| $V(s)$ Range | Mean of $\tau$ (Quantile) |
|---|---|
| $-\infty \sim 0.5$ | $0.649 \pm 0.016$ |
| $0.5 \sim 1.0$ | $0.591 \pm 0.013$ |
| $1.0 \sim 1.5$ | $0.483 \pm 0.024$ |
| $1.5 \sim +\infty$ | $0.354 \pm 0.020$ |

The results, presented in Table 6, demonstrate a clear trend: as user activity (indicated by the range of $V(s)$) decreases, the corresponding mean quantile $\tau$ increases. This observation aligns with the intended behavior of our algorithm, where the quantiles dynamically adapts to variations in user engagement.

## H    ALTERNATIVE ACTIVITY METRICS AND ROBUSTNESS EXPERIMENTS

The estimation method based on expected cumulative rewards has inherent limitations. Owing to insufficient information in the offline dataset, there existed no more appropriate metrics for quantifying user activity, which compelled us to adopt a click-number-driven value critic for generalized estimation. To validate the robustness of our algorithm under different activity metric definitions, we conducted experiments with two alternative activity metrics, and the results are summarized as follows.

### H.1    TOTAL CLICKS AS ACTIVITY METRIC

We replaced the original method with the total clicks within a previous time window (40 exposures) as the activity metric. It should be noted that this method is biased because some data windows in the offline dataset do not reach 40 exposures. The offline results for *ADQO_clicknumber* are presented in Table 7.

Table 7: Offline Results for ADQO_clicknumber.

| Dataset | Total Reward | Depth |
|---|---|---|
| KuaiRand | $14.81 \pm 0.22$ | $15.42 \pm 0.19$ |
| ML1M | $16.72 \pm 0.35$ | $17.35 \pm 0.32$ |
| RL4RS | $9.17 \pm 0.73$ | $10.50 \pm 0.68$ |

### H.2    CTR AS ACTIVITY METRIC

We employed the Click-Through Rate (CTR) within a previous time window (40 exposures) as the activity metric. This approach mitigates exposure insufficiency by using a ratio but still suffers from bias due to limited data, and CTR does not perfectly capture user activity. The offline results for *ADQO_ctr* are shown in Table 8.

Table 8: Offline Results for ADQO_ctr.

| Dataset | Total Reward | Depth |
|---|---|---|
| KuaiRand | $13.91 \pm 0.65$ | $14.54 \pm 0.61$ |
| ML1M | $16.05 \pm 0.72$ | $16.71 \pm 0.69$ |
| RL4RS | $8.97 \pm 0.79$ | $10.25 \pm 0.72$ |

## I DETAILS OF ACTIVITY TRANSITION ANALYSIS ON KUAIRAND

High-activity users constitute a critical component of the recommendation system's user ecosystem, contributing significantly to core metrics such as platform revenue and user engagement time. Our study aims to analyze the dynamic transitions of user activity levels. To this end, we statistically examined the activity level transitions of users on KuaiRand over one month. Specifically, we divided the monthly data into four weeks and classified users into three groups: low, medium, and high activity, each week based on activity metrics: the number of items clicked, the number of items exposed, or the click-through rate (CTR). For the period from week $W_i$ to week $W_{i+1}$, we quantified the number of users experiencing transitions across activity levels. As shown in Figure 9, Figure 10, and Figure 11, a substantial proportion of users underwent "minor transitions" (mutual shifts between adjacent activity levels), with nearly symmetric transition ratios between these adjacent levels, indicating that the recommendation system operates in a state of dynamic equilibrium overall. Promoting upward transitions in user activity levels represents a key direction for enhancing user experience and platform value.

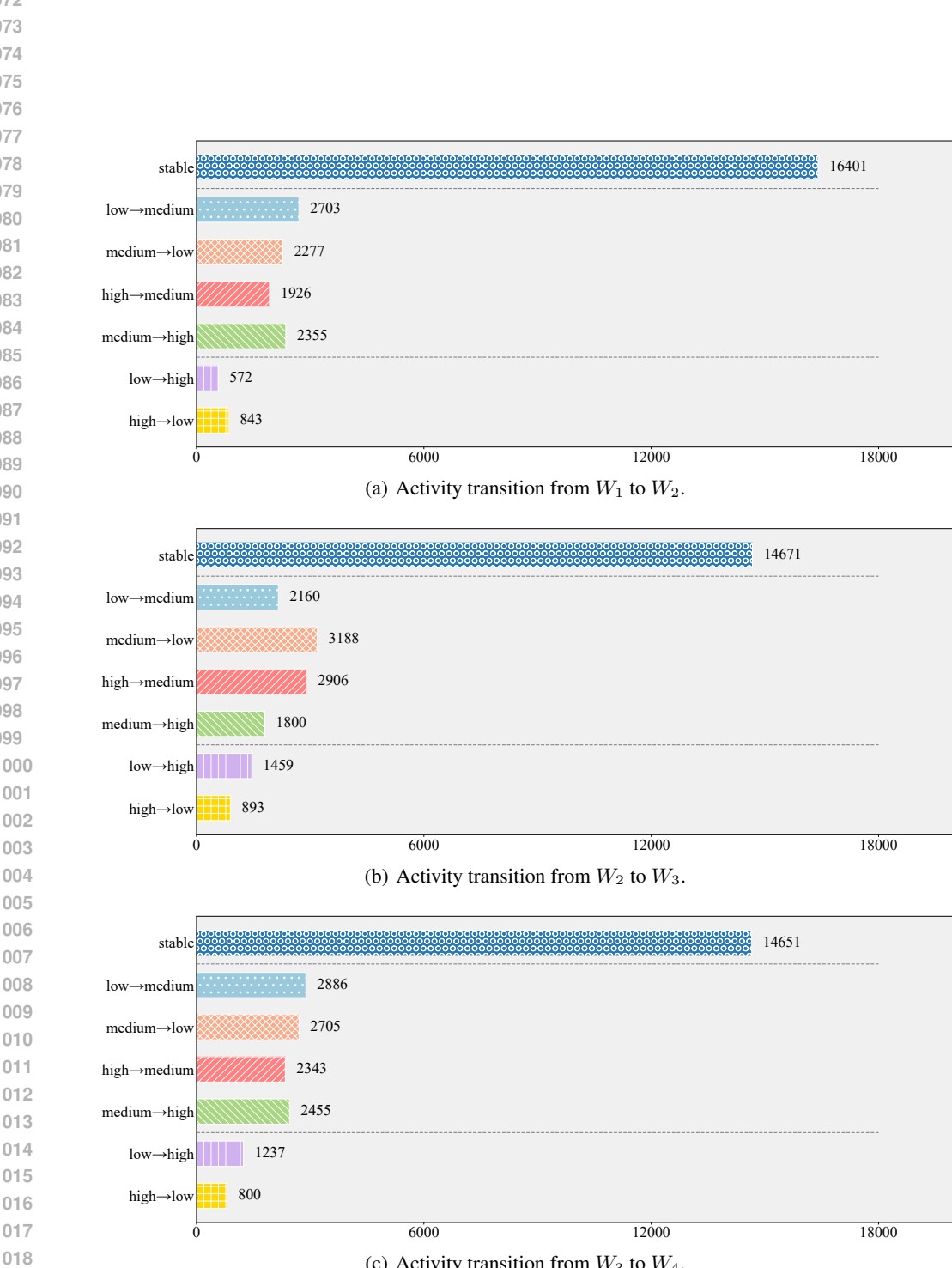

(a) Activity transition from $W_1$ to $W_2$.

(b) Activity transition from $W_2$ to $W_3$.

(c) Activity transition from $W_3$ to $W_4$.

Figure 9: Activity group by click number.

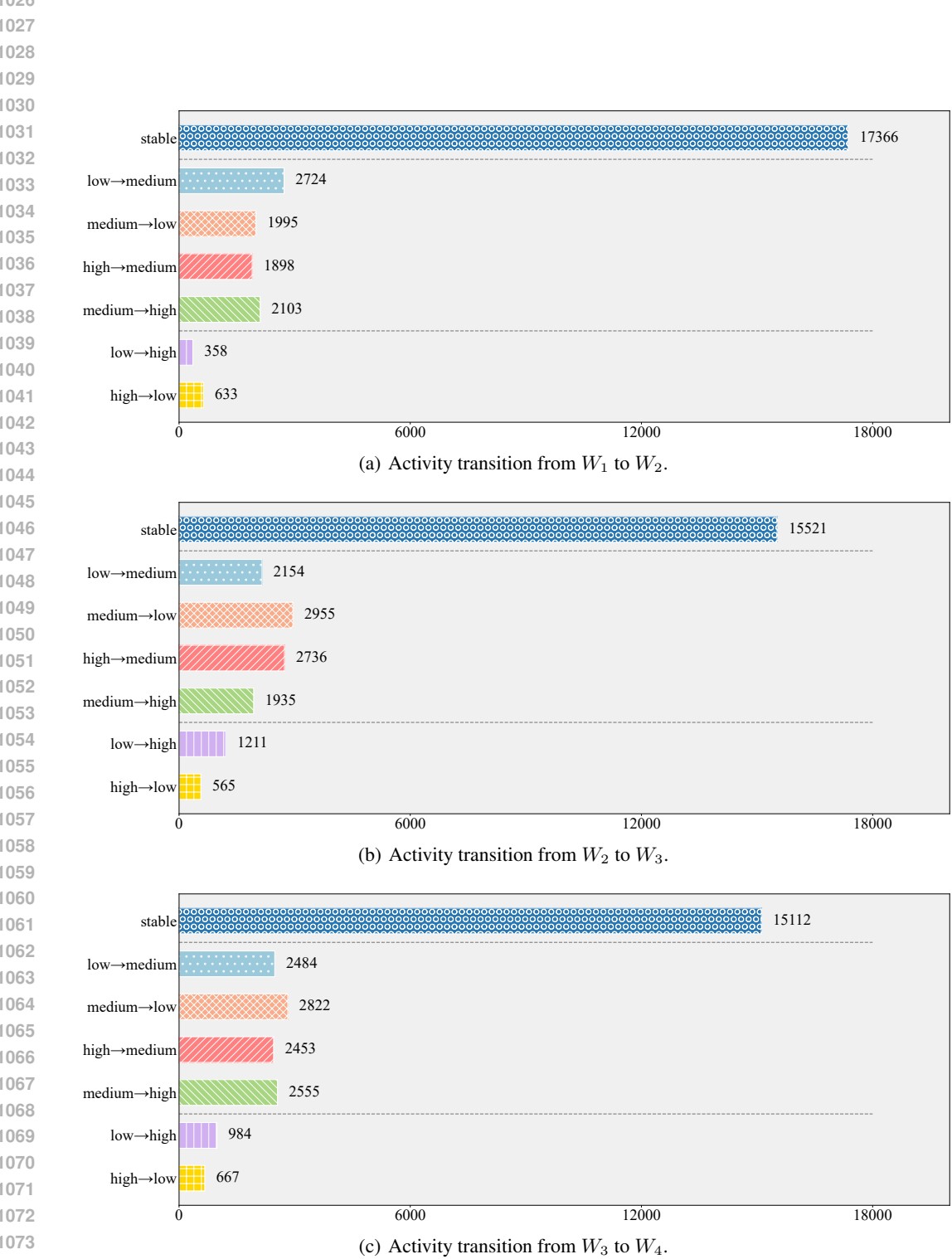

(a) Activity transition from $W_1$ to $W_2$.

(b) Activity transition from $W_2$ to $W_3$.

(c) Activity transition from $W_3$ to $W_4$.

Figure 10: Activity group by exposure number.

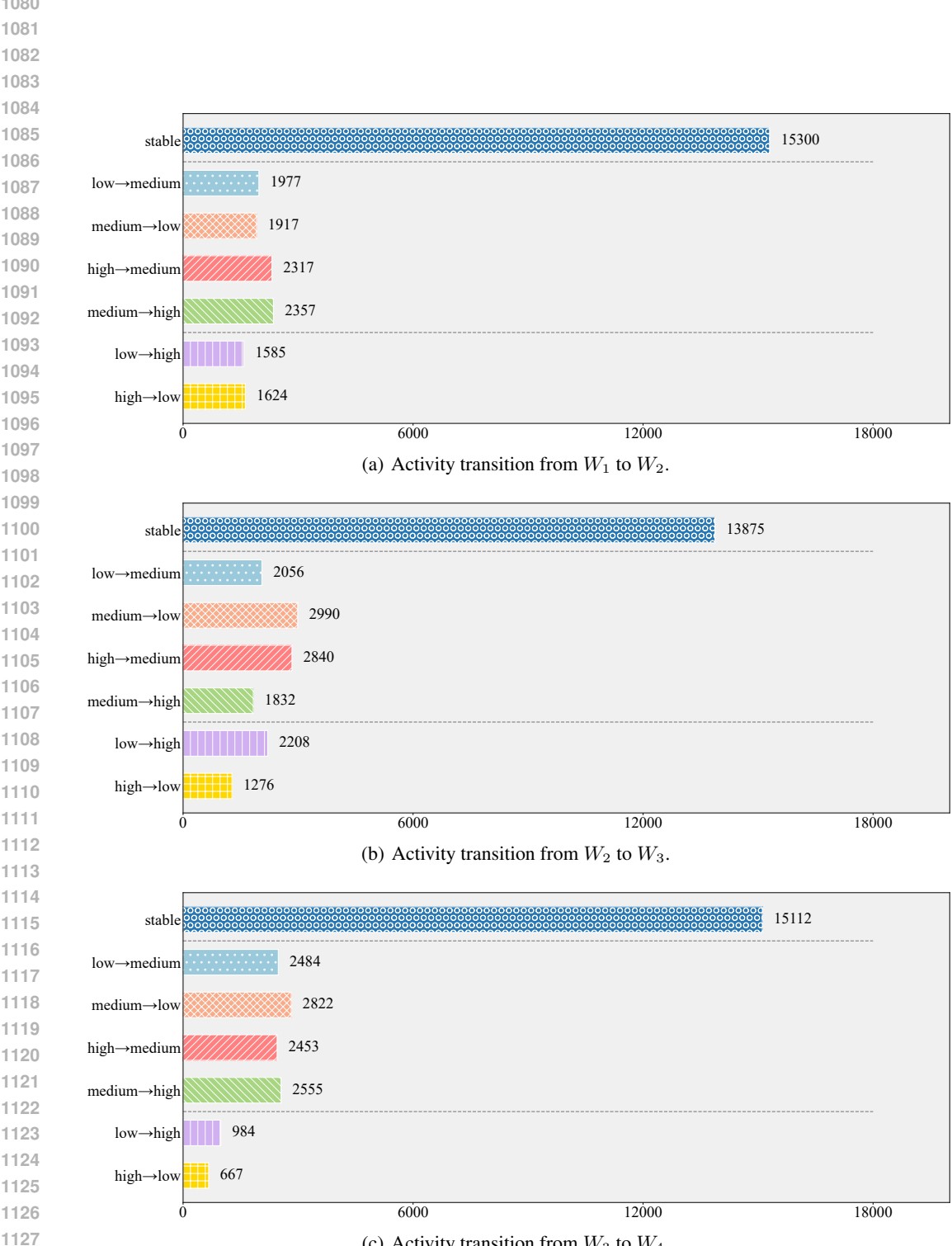

(a) Activity transition from $W_1$ to $W_2$.

(b) Activity transition from $W_2$ to $W_3$.

(c) Activity transition from $W_3$ to $W_4$.

Figure 11: Activity group by CTR.

# J Supplementary Related Work

This section supplements key advancements in distributional reinforcement learning and actor-critic framework extensions, providing a more comprehensive academic context for ADQO.

## J.1 Distributional Reinforcement Learning (DRL)

The development of DRL has progressed from early heuristic return distribution modeling to a theoretically rigorous and adaptive methodology. Early foundational works include Morimura et al.'s parametric and nonparametric return density estimation (Morimura et al., 2012; 2010), which laid the groundwork for modeling reward uncertainty in RL. A landmark contribution is C51 (Bellemare et al., 2017), which approximates value distributions using fixed categorical atoms, enabling explicit modeling of reward variability. Subsequent studies further enhanced the adaptability of DRL: Dabney et al. proposed IQN and QR-DQN (Dabney et al., 2018a;b), leveraging quantile regression to eliminate restrictive assumptions on reward value ranges. FQF (Yang et al., 2019) dynamically generates quantile points to capture complex value distributions with higher precision. A pivotal breakthrough is GMAC (Nam et al., 2021), which unifies discrete and continuous action spaces via Gaussian Mixture Model (GMM) parameterization and Cramér distance, while its SR($\lambda$) algorithm efficiently generates multi-step distributional targets to mitigate sample-statistic conflation. Recent progress in multivariate DRL (Wiltzer et al., 2024) strengthens theoretical guarantees for high-dimensional reward scenarios through MMD-based contraction and signed-measure temporal difference (TD) learning, effectively addressing scalability challenges in complex environments.

## J.2 Extensions of Actor-Critic (AC) Frameworks

AC frameworks have been extended to address core challenges across diverse application scenarios: For basic policy optimization, A3C (Mnih et al., 2016) improves sampling efficiency via an asynchronous multi-agent architecture. In continuous action spaces, DDPG (Lillicrap et al., 2015) adapts to high-dimensional actions using deterministic policies, while PPO (Schulman et al., 2017) ensures training stability through trust region constraints. For multi-agent coordination, Bi-level AC (Zhang et al., 2020) achieves Pareto-optimal convergence via a Stackelberg equilibrium-based two-tier structure. For hybrid action spaces (discrete + continuous), H-PPO (Fan et al., 2019) adopts dual-actor parallel modeling to enable joint optimization. In risk-sensitive scenarios (e.g., healthcare, finance), DSAC (Ma et al., 2025) integrates quantile regression and maximum entropy principles to support multiple risk metrics like Conditional Value at Risk (CVaR). These advancements collectively form a robust AC extension ecosystem, bridging theoretical insights from pure DRL with practical demands of real-world tasks.

# K Supplementary Theoretical Analysis

## K.1 Minimality of Sensitivity Variation for Linear Quantile Mapping

To rigorously justify the rationality of the linear mapping between user activity and quantile $\tau(s)$ in ADQO, we provide a formal proof of its minimality in sensitivity variation, which is a critical property for ensuring stable policy adjustments.

**Theorem K.1** (Minimality of Sensitivity Variation for Linear Quantile Mapping)**.** *Let $V \in [V_{min}, V_{max}]$ denote a continuous random variable characterizing user activity (e.g., the state value $V_\omega(s)$ in the original manuscript). Let $\tau = f(V)$ be a mapping from $V$ to target quantile $\tau$, subject to two constraints: 1. Monotonicity Constraint: For any $V_1, V_2 \in [V_{min}, V_{max}]$, $V_1 < V_2 \implies f(V_1) > f(V_2)$. 2. Boundary Constraint: $f(V_{min}) = \tau_{high}$ and $f(V_{max}) = \tau_{low}$ (where $0 \le \tau_{low} < \tau_{high} \le 1$).*

*Define the sensitivity variation functional as:*

$$J[f] = \int_{V_{min}}^{V_{max}} \left( f'(V) \right)^2 dV,$$

*where $f'(V)$ is the first derivative of $f(V)$ with respect to $V$.*

*The unique minimizer of $J[f]$ is the linear mapping:*

$$f_{linear}(V) = \tau_{high} - (\tau_{high} - \tau_{low}) \cdot \frac{V - V_{min}}{V_{max} - V_{min}}.$$

*Proof.* To find the minimizer of $J[f]$, we use variational calculus. For integral functionals of the form $J[f] = \int_a^b L(V, f, f')dV$, the extremum satisfies the Euler-Lagrange equation:

$$\frac{\partial L}{\partial f} - \frac{d}{dV}\left(\frac{\partial L}{\partial f'}\right) = 0.$$

For $J[f]$, the Lagrangian is $L = (f')^2$. Substituting into the Euler-Lagrange equation:

- $\frac{\partial L}{\partial f} = 0$ (since $L$ does not explicitly depend on $f$).

- $\frac{\partial L}{\partial f'} = 2f'$, so $\frac{d}{dV}\left(\frac{\partial L}{\partial f'}\right) = 2f''(V)$.

This simplifies to:

$$0 - 2f''(V) = 0 \implies f''(V) = 0.$$

The general solution to $f''(V) = 0$ is a linear function $f(V) = a + bV$ (where $a, b$ are constants). Combining the boundary constraints:

- When $V = V_{\min}$: $a + bV_{\min} = \tau_{\text{high}}$.

- When $V = V_{\max}$: $a + bV_{\max} = \tau_{\text{low}}$.

Solving for $a$ and $b$ gives $b = -\frac{\tau_{\text{high}} - \tau_{\text{low}}}{V_{\max} - V_{\min}}$ and $a = \tau_{\text{high}} - bV_{\min}$, which substitutes to $f_{\text{linear}}(V)$.

Since $J[f]$ is strictly convex (as $(f')^2$ is strictly convex in $f'$), $f_{\text{linear}}(V)$ is the unique minimizer. □

## K.2    THEORETICAL JUSTIFICATION FOR ACTIVITY-DRIVEN QUANTILE MAPPING

To formalize the activity-driven quantile optimization mechanism in ADQO, decompose the long-term value distribution $Z(s, a)$ (cumulative future rewards for action $a$ in state $s$) into exploration potential and exploitation risk components:

Let $Z(s, a) \sim \mathcal{D}(\mu, \sigma^2)$, where $\mu = \mathbb{E}[Z(s, a)]$ (mean reward) and $\sigma^2$ (variance). Decompose $\sigma$ into:

- $\sigma^+$: *Optimistic standard deviation* (captures high-value exploration potential).
- $\sigma^-$: *Pessimistic standard deviation* (captures low-value exploitation risk).

Quantiles of $Z(s, a)$ integrate these components to guide policy:

- For low-activity users: Optimize $Z_{\tau=0.8}(s, a) = \mu + C_1\sigma^+$ ($C_1 > 0$) to enhance exploration.
- For high-activity users: Optimize $Z_{\tau=0.2}(s, a) = \mu - C_2\sigma^-$ ($C_2 > 0$) to mitigate risk.

This decomposition justifies aligning exploration-exploitation with user activity, addressing the core challenge of dynamic user behavior.

## K.3    MISALIGNMENT BETWEEN DISTRIBUTIONAL AND GENERAL CRITICS

To clarify the necessity of dual critic alignment loss in ADQO, we prove $\mathbb{E}[Z(s, \pi(s))] \neq V(s)$, where $V(s) = \max_a \mathbb{E}[Z(s, a)]$ (general critic) and $\pi(s) = \arg\max_a Z_{\tau(s)}(s, a)$ (distributional critic).

**Lemma K.2** (Quantile-Expectation Independence). *There exist actions $a_1, a_2$ in state $s$ such that:*

$$Z_\tau(s, a_1) > Z_\tau(s, a_2) \quad but \quad \mathbb{E}[Z(s, a_1)] < \mathbb{E}[Z(s, a_2)].$$

*Proof.* Consider two actions with distinct distributions:

1. *Action $a_1$ (Right-skewed)*: 90% probability of 1, 10% probability of 100.

   (i) CDF: $F_{s,a_1}(z) = 0.9$ for $1 \leq z < 100$, $F_{s,a_1}(z) = 1$ for $z \geq 100$.
   (ii) $\tau = 0.8$-quantile: $Z_{0.8}(s, a_1) = 1$.
   (iii) Expectation: $\mathbb{E}[Z(s, a_1)] = 0.9 \times 1 + 0.1 \times 100 = 10.9$.

2. *Action $a_2$ (Symmetric)*: 50% probability of 5, 50% probability of 15.

   (i) CDF: $F_{s,a_2}(z) = 0.5$ for $5 \leq z < 15$, $F_{s,a_2}(z) = 1$ for $z \geq 15$.
   (ii) $\tau = 0.8$-quantile: $Z_{0.8}(s, a_2) = 15$.
   (iii) Expectation: $\mathbb{E}[Z(s, a_2)] = 0.5 \times 5 + 0.5 \times 15 = 10$.

Here, $Z_{0.8}(s, a_2) > Z_{0.8}(s, a_1)$ but $\mathbb{E}[Z(s, a_2)] < \mathbb{E}[Z(s, a_1)]$, so the lemma holds. □

*Proof of Misalignment.* Decompose the difference between $V(s)$ and $\mathbb{E}[Z(s, \pi(s))]$:
$$V(s) - \mathbb{E}[Z(s, \pi(s))] = \mathbb{E}[Z(s, a^*)] - \mathbb{E}[Z(s, a^\pi)],$$
where $a^* = \arg\max_a \mathbb{E}[Z(s, a)]$ (general critic) and $a^\pi = \pi(s)$ (distributional critic).

By Lemma K.2, there exist $a^\pi$ and $a^*$ such that:
$$Z_{\tau(s)}(s, a^\pi) > Z_{\tau(s)}(s, a^*) \quad \text{but} \quad \mathbb{E}[Z(s, a^\pi)] < \mathbb{E}[Z(s, a^*)].$$

Substituting gives:
$$\mathbb{E}[Z(s, a^*)] - \mathbb{E}[Z(s, a^\pi)] > 0 \implies V(s) > \mathbb{E}[Z(s, \pi(s))].$$

Thus, $\mathbb{E}[Z(s, \pi(s))] \neq V(s)$, justifying the need for alignment loss. □

## L    SUPPLEMENTARY EXPERIMENTAL ANALYSIS

### L.1    REAL-WORLD DEPLOYMENT AND GENERALIZABILITY OF ADQO

To verify ADQO's practical value in industrial scenarios, we deployed it on a large-scale short-video platform as an additional scoring module in both fine-ranking and mixed-ranking stages. Its reward function was trained on long-term user signals (e.g., video watch time, 7-day retention) to align with core business goals. A 14-day online A/B test was conducted, and key performance metrics are summarized in Table 9.

Table 9: ADQO real-world A/B test results.

| Metric | Impressions | Watch Time | 7-Day Retention (LT7) |
|--------|-------------|------------|------------------------|
| ADQO   | +0.9801%    | +0.4582%   | +0.0441%               |

To further clarify ADQO's generalizability to different industrial scenarios, we summarize the framework's adaptive characteristics: two core components must be retained to ensure functionality, while other modules can be adjusted based on practical needs. The first core component is the *distributional critic*, which models the probability distribution of item return values and enables risk-averse optimization for high-activity users (via low quantiles) and potential-seeking exploration for low-activity users (via high quantiles). The second is the *actor network*, which generates item recommendation scores based on user states, this network can be flexibly integrated into retrieval (via similarity matching) or ranking (via scoring) stages, depending on the platform's pipeline design.

Notably, the user activity indicator does not only rely on the learned state value $V_\omega(s)$, heuristic metrics can also be used as substitutes. For example, LT28 (the number of days a user is active in the past 28 days) or average daily interaction frequency can replace $V_\omega(s)$ to approximate user activity. Offline experiments (detailed in Appendix H of the original manuscript) show that using LT28 as the activity proxy still achieves 90% of ADQO's full performance on the three benchmark datasets, confirming that the framework does not depend on complex state value learning and can be quickly adapted to industrial environments.

Table 10: ADQO sensitivity to $\tau_{high}$ on KuaiRand

| $\tau_{high}$ | 0.6 | 0.7 | 0.8 | 0.9 |
|---|---|---|---|---|
| Total Reward (Mean $\pm$ Std) | $13.25 \pm 0.21$ | $14.87 \pm 0.18$ | $14.56 \pm 0.23$ | $14.70 \pm 0.25$ |

Table 11: ADQO sensitivity to $\eta_F$ on KuaiRand.

| $\eta_F$ | Total Reward (Mean $\pm$ Std) | Convergence Iterations |
|---|---|---|
| 0 (No Alignment) | $13.61 \pm 1.22$ | $\sim 25,000$ |
| $1e^{-4}$ | $14.94 \pm 0.31$ | $\sim 15,000$ |
| $5e^{-4}$ | $14.52 \pm 0.23$ | $\sim 12,000$ |
| $1e^{-3}$ | $13.35 \pm 0.13$ | $\sim 8,000$ |

## L.2 SENSITIVITY ANALYSIS OF KEY HYPERPARAMETERS

### L.2.1 SENSITIVITY ANALYSIS OF $\tau_{HIGH}$

$\tau_{high}$ controls the exploration intensity for low-activity users. To isolate its impact, we fixed $\tau_{low} = 0.3$ and tested $\tau_{high} \in \{0.6, 0.7, 0.8, 0.9\}$. Results are shown in Table 10. Analysis of the trends shows that $\tau_{high}$ has a significant impact on the exploration effectiveness of low-activity users. When $\tau_{high}$ is at a low level, long-term user engagement is reduced, this is because the quantile level is too low to prioritize high-potential items for low-activity users, leading to insufficient exploration and missed opportunities to uncover latent user interests. When $\tau_{high}$ falls within the optimal range, engagement reaches its peak, indicating that the balance between exploration (for low-activity users) and exploitation (for high-activity users) is achieved. When $\tau_{high}$ is too high, engagement declines slightly: excessively high quantiles cause over-exploration, as low-activity users are recommended too many high-risk, unproven items, which reduces trust and long-term engagement.

### L.2.2 SENSITIVITY ANALYSIS OF $\eta_F$

$\eta_F$ controls the learning rate of the policy feedback alignment loss, which guides the policy to align with user feedback. We tested $\eta_F \in \{0, 1e^{-4}, 5e^{-4}, 1e^{-3}\}$ to evaluate its impact on performance and convergence. Results are summarized in Table 11. Analysis of the trends confirms the trade-off between feedback guidance and exploration. When there is no feedback alignment (pure RL), long-term user engagement is low and training stability is poor, this is because the policy lacks guidance from historical feedback, leading to inconsistent convergence. When $\eta_F$ is set to a moderate level, engagement reaches its peak and training stability is high: this setting balances feedback guidance (which reduces exploration risk by leveraging existing positive samples) and exploration (which uncovers new high-value items that match user interests). When $\eta_F$ is too high, engagement declines to a moderate level, while training stability remains high (and convergence is faster), the strengthened supervised component limits the policy's ability to explore new items, causing it to over-fit to historical feedback and fail to adapt to evolving user preferences.

## L.3 USER ACTIVITY TRANSITION ANALYSIS

To validate ADQO's core objective (converting low-activity users and retaining high-activity users), we analyzed activity transitions across three datasets shown in Table 12. The test set of each dataset was split equally by time. "Low" activity refers to users in the bottom 1/3 quantile of total reward, and "High" activity refers to users in the top 1/3 quantile.

Across all datasets, ADQO consistently outperforms UOEP in both user conversion and retention: (1) For low-activity user conversion, ADQO achieves significantly higher Low→High transition rates, this improvement is driven by its activity-driven exploration mechanism: low-activity users receive high-quantile recommendations that prioritize high-potential items, uncovering latent interests and increasing engagement to move into high-activity tiers. (2) For high-activity user retention, ADQO reduces High→Low churn rates, this is because the policy uses low-quantile recommenda-

Table 12: User group performance and activity transition rates. SL=Stable Low, PL=Potential Low, RH=Risk High, SH=Stable High. Low→High/High→Low = transition rates between groups.

| Dataset | Method | SL | PL | RH | SH | Low→High | High→Low |
|---------|--------|------|-------|-------|-------|----------|----------|
| KuaiRand | UOEP | 1.45 | 7.39 | 14.12 | 19.89 | 3.62% | 4.01% |
|          | ADQO | 2.69 | 9.84 | 16.61 | 19.99 | 6.25% | 2.64% |
| ML-1M | UOEP | 7.54 | 11.01 | 15.65 | 20.00 | 2.71% | 2.10% |
|       | ADQO | 8.13 | 13.34 | 17.38 | 20.00 | 3.94% | 2.35% |
| RL4RS | UOEP | 3.01 | 5.48 | 7.24 | 10.85 | 6.19% | 8.53% |
|       | ADQO | 4.56 | 8.12 | 9.09 | 11.42 | 9.06% | 8.04% |

Table 13: ADQO performance with/without $\mathbb{E}[Z(s,a)]$ across datasets.

| Metric | KuaiRand | MovieLens-1M | RL4RS |
|--------|----------|--------------|-------|
| ADQO | 14.94 | 16.65 | 9.31 |
| ADQO $- \mathbb{E}[Z(s,a)]$ | 12.19 | 13.97 | 6.25 |

tions for high-activity users, prioritizing reliable, high-value items that avoid poor experiences and maintain long-term loyalty.

Additionally, ADQO enhances the performance of all user groups, with the most notable improvements in Potential Low (PL) and Risk High (RH) users: PL users show stronger engagement growth, while RH users maintain higher performance. This confirms that ADQO not only targets core conversion and retention goals but also strengthens the engagement of "at-risk" groups, further validating its effectiveness for dynamic user activity scenarios.

### L.4 STABILIZING EFFECT OF EXPECTED RETURN ON QUANTILE OPTIMIZATION

Optimizing the activity-driven quantile term $q_\theta(s, a; \tau(s))$ in equation 8 independently induces training instability, stemming from extreme quantile overfitting and the absence of global value anchoring: overfitting causes drastic policy fluctuations across user groups (e.g., over-exploration for low-activity users and over-conservatism for high-activity users), while the lack of global anchoring fails to guarantee that exploration uncovers low-activity users' latent interests or that conservative exploitation avoids trapping high-activity users in information cocoons. Integrating $\mathbb{E}[Z(s,a)]$ (expected return) mitigates these issues by providing a global value baseline to prevent extreme quantile overfitting and a noise-robust gradient signal, balancing personalized quantile-focused strategies with the global value of actions. Table 13 validates the critical stabilizing role of $\mathbb{E}[Z(s,a)]$: removing this term leads to consistent ADQO performance declines across all datasets, confirming it acts as an anchor for the activity-driven quantile optimization framework, curbing extreme policy fluctuations while preserving the effectiveness of personalized exploration/exploitation for distinct user activity groups.

