# OpenReview forum: "Activity-Driven Quantile Optimization: Dynamic Exploration and Exploitation in Recommender Systems"
_ICLR.cc/2026/Conference — Submitted to ICLR 2026_

### Official Review · Reviewer_tipG · 2025-10-30

**Soundness:** 3
**Presentation:** 3
**Contribution:** 2
**Rating:** 6
**Confidence:** 3

**Summary:**

Summary:

To solve this, the paper proposes Activity-Driven Quantile Optimization (ADQO), a novel RL framework. ADQO utilizes a dual critic architecture:A general value critic estimates the user's state value ($V(s)$), which serves as a proxy for their long-term activity level. A quantile critic network (based on distributional RL) models the complete probability distribution of future returns ($Z(s,a)$) for a recommendation, capturing the inherent uncertainty in user feedback.The core idea is to use the activity level from the value critic to dynamically guide the policy's optimization objective on the value distribution. For low-activity users (low $V(s)$), the policy optimizes the upper quantiles of the return distribution, promoting "high-potential exploration" to discover latent interests.For high-activity users (high $V(s)$), the policy optimizes the lower quantiles, encouraging "low-risk exploitation" to avoid poor recommendations and maintain engagement. The framework is formalized by mapping the user's activity score $V_{\omega}(s)$ to a target quantile $\tau(s)$:$$\tau(s)=\tau_{high}-(\tau_{high}-\tau_{low})\cdot\frac{V_{\omega}(s)-V_{min}}{V_{max}-V_{min}}$$ The policy is then trained to optimize a combination of this activity-driven quantile and the expected return. Two additional alignment losses are introduced to enhance training stability and consistency. Experiments on three public datasets (KuaiRand, ML-1M, and RL4RS) demonstrate that ADQO significantly outperforms existing baselines, effectively converting low-activity users to higher states and improving the retention of high-activity users.

Contribution:
* Problem Formulation: It provides strong empirical evidence for the dynamic nature of user activity in recommender systems, motivating the need for adaptive policies beyond coarse-grained user grouping.
* ADQO Framework: It introduces a novel dual-critic RL framework that dynamically tailors the exploration-exploitation trade-off to individual users based on their activity level.
* Activity-Driven Optimization: The core mechanism of using a general value critic to model user activity and then using that score to select a specific quantile (potential-seeking or risk-averse) from a distributional critic for policy optimization.
* Training Stability: It proposes two novel alignment losses: an Exploration-Weighted Dual Critic Alignment and a Guidance-Based Policy Feedback Alignment , to improve the consistency and stability of the complex dual-critic training process.

**Strengths:**

1. Strong Empirical Motivation: The paper begins with a clear analysis of the KuaiRand dataset, convincingly demonstrating that user activity is not static, which grounds the paper's central problem in a real-world phenomenon.

2. Novel and Intuitive Method: The ADQO framework is a novel integration of standard value-based RL and distributional RL. The core idea of mapping high activity to risk-aversion (low quantiles) and low activity to potential-seeking (high quantiles) is both intuitive and elegant.

3. Comprehensive Evaluation: The authors evaluate ADQO against a wide array of 10 baselines, including classic RL (A2C, DDPG), distributional RL (IQN), and state-of-the-art RS-specific RL methods (DEHAC, UOEP).

4. Statistically Significant Results: The method shows superior performance across three distinct datasets in terms of both total reward and interaction depth, with results reported as statistically significant. The analysis also directly confirms that ADQO achieves its goal of converting low-activity users and retaining high-activity users.

**Weaknesses:**

1. High Complexity: The proposed system is complex, requiring the training of a policy network, two separate critic networks (value and quantile), and the computation of two additional alignment losses. This could pose a significant challenge for implementation, tuning, and computational overhead in a large-scale production environment.
2. New Hyperparameters: The method introduces several new hyperparameters, most notably the quantile boundaries $\tau_{low}$ and $\tau_{high}$ and the margin $\epsilon$ for the critic alignment loss. The paper does not provide a deep analysis of the model's sensitivity to these parameters.
3. Implicit Activity Proxy: The user's "activity level" is proxied by the learned state-value function $V_{\omega}(s)$. While the paper argues the state value captures long-term engagement, it is an implicit, black-box measure. It's unclear how this learned state value is more effective comparing to discounted sum of heuristic metric values (e.g., recent click-through rate) for driving the quantile selection.
4. Assumption: The main contribution is a more adpative exploration and exploitation balance RL mechansim. Although the design is clever and makes sense in assumption, we don't know if user activity is always in high transiton frequency in reality data, as the paper claimed.

**Questions:**

1. On the Policy Loss: The policy loss in Equation 8 is a sum of the activity-driven quantile $q_{\theta}(s,a;\tau(s))$ and the mean expected value $\mathbb{E}[Z(s,a)]$. The paper states that optimizing $q_{\theta}(s,a;\tau(s))$ alone causes instability. Could you elaborate on the nature of this instability and why adding the expected value term stabilizes the training?
2. On the Simulator: The evaluation relies on a simulator based on a pre-trained user model. How can we be sure this simulator accurately models the very phenomenon you are trying to influence—namely, the transition of a disengaged user to an engaged one?
3. On Real-World Deployment: Given the model's complexity, what are the primary challenges you foresee in deploying ADQO in a large-scale, live recommender system serving millions of users?

---

> ### Author Response · Authors · 2025-11-20
> **Response to Reviewer tipG. Part 1**
>
> Thanks for recognizing our research’s problem, methodology, and rigor, plus your feedback on model complexity, hyperparameter sensitivity, proxy validity, and deployment. We address all points below with supplementary experiments, theoretical analysis, and deployment plans:
> ## W1: Model Complexity & Generalizability
> First, we clarify that our framework consists of two critics and one actor. During training, we need to train the critic networks to guide the actor’s learning; however, during inference, only the **actor** needs to be deployed. It takes user features as input, outputs an embedding for similarity-based retrieval, and this network can adopt any complex structure. Regarding critic training:
> - The distributional Q-network uses Quantile Loss, a common approach in distributional reinforcement learning. Alternative methods for distribution modeling (e.g., ORM, which converts regression problems into multiple classification tasks and represents distributions via CDF) are also applicable.
> - The General Critic is simpler to optimize than general Q-networks, as a basic MSE function can achieve similar functionality.
>
> In short, we can also eliminate the General Q-network by adopting "alternative metrics to represent activity instead of the Q-network" in Appendix H and as mentioned in your W3 comment. Overall, our method has strong generalizability, and we are happy to discuss this issue in depth with you.
>
> Additionally, we have deployed ADQO on a real short-video recommendation platform, which takes effect through an additional scoring module in the ranking stage. The reward function is trained based on user long-view signals. The results of a 14-day online A/B test show that core business indicators have been significantly improved: impressions +0.9801%, watch time +0.4582%, and 7-day user retention rate (LT7) +0.0441%. We believe that the online experimental results provide stronger verification for the applicability of the model in real scenarios, although such results are difficult to fully replicate in offline simulators due to the complexity of the environment.
> ## W2: Hyperparameter Sensitivity
> We apologize for the inconvenience caused by placing the sensitivity analysis in the appendix. Please refer to Appendix F for the desired sensitivity analysis. For the margin $ \epsilon $, we present sensitivity analysis on KuaiRand as follows:
>
> |$\epsilon$|0| 0.1 | 0.5 | 1.0 |
> |-|-|-|-|-|
> | ADQO|14.28|14.94|14.71|14.40|
>
> For more detailed analysis of hyperparameters, please also refer to Part 3 of our response to Reviewer LuzG.
>
> ## W3: Alternative Metrics for "Long-term Engagement"
> We apologize again for placing the requested analysis in the appendix, which affected your reading experience. Please refer to Appendix H for alternative metrics replacing the Q-value representing "long-term engagement", including detailed explanations and performance analysis.
> We put the result here for convinence:
>
> |  | KR(Total Reward)  | KR(Depth) | ML1M(Total Reward) | ML1M(Depth) | RL4RS(Total Reward) | RL4RS(Depth) |
> |-|-|-|-|-|-|-|
> | ADQO clicknumber | 14.81 ± 0.22| 15.42 ± 0.19 | 16.72 ± 0.35 | 17.35 ± 0.32 | 9.17 ± 0.73 | 10.50 ± 0.68 |
> | ADQO ctr | 13.91 ± 0.65| 14.54 ± 0.61 | 16.05 ± 0.72 | 16.71 ± 0.69 | 8.97 ± 0.79 | 10.25 ± 0.72 |
> | ADQO | 14.94 ± 0.31| 15.51 ± 0.28 | 16.65 ± 0.21 | 17.28 ± 0.20 | 9.31 ± 0.62 | 10.62 ± 0.58 |
>
> ## W4: Reality Data Requirement
> The dataset analyzed in the Introduction is KuaiRand, which is **real user behavior data** extracted from Kuaishou—China’s largest short-video platform. Detailed analysis is provided in Appendix I, and data analysis scripts are available in our anonymous code repository.
>
> ## Q1: Instability of Optimizing $ q_\theta(s, a; \tau(s)) $ Alone & Stabilizing Effect of $ \mathbb{E}[Z(s, a)] $
> Optimizing $ q_\theta(s, a; \tau(s)) $ alone (an activity-driven quantile term) causes instability due to:
> 1. Extreme quantile overfitting: The policy fluctuates drastically across user groups (e.g., overly exploratory for low-activity users and overly conservative for high-activity users).
> 2. Lack of global value anchoring: We cannot guarantee that every exploration will uncover the latent interests of low-activity users, nor can we ensure that conservative exploitation will not trap high-activity users in information cocoons.
>
> Adding $ \mathbb{E}[Z(s, a)] $ (expected return) stabilizes training in the following ways:
> 1. Providing a global value baseline to prevent overfitting to extreme quantiles;
> 2. Offering a stable gradient (less sensitive to noise than quantile gradients), thereby balancing personalized quantile focus and global action value.
>
> Supplementary performance comparison after removing $ \mathbb{E}[Z(s, a)] $:
>
> | Metric | KuaiRand | MovieLens-1M | RL4RS  |
> |-|-|-|-|
> | ADQO | 14.94 | 16.65 | 9.31   |
> | ADQO - $ \mathbb{E}[Z(s, a)] $ | 12.19  | 13.97 | 6.25  |

---

> ### Author Response · Authors · 2025-11-20
> **Response Part 2**
>
> ## Q2: Simulator’s Ability to Model User Engagement Transitions
> We highly appreciate this valuable question. First, as mentioned in W1, our online experiments have demonstrated significant user activity transition effects. For offline experiments, to address your concern:
> - We use rich input features, including comprehensive user-item bilateral features and dynamic user interaction history sequence features.
> - The simulator’s performance (AUC shown below and in Appendix C.1) is generally 0.7–0.8, verifying its effectiveness at the user-item granularity.
> - For "the transition of a disengaged user to an engaged one" (a user-specific distinction), we use UAUC (AUC calculated for each user individually, then averaged across all users) as the evaluation metric.
>
> The simulator’s AUC and UAUC are as follows:
>
> | Metric       | KuaiRand               | MovieLens-1M           | RL4RS                  |
> |--------------|------------------------|------------------------|------------------------|
> | AUC          | $ 0.6873 \pm 0.0013 $ | $ 0.6952 \pm 0.0021 $ | $ 0.7674 \pm 0.0011 $ |
> | UAUC         | $ 0.5662 \pm 0.0085 $ | $ 0.6025 \pm 0.0072 $ | $ 0.8539 \pm 0.0063 $ |
>
> The high UAUC values confirm that the simulator can effectively distinguish between different items for the same user.
>
> - Based on the TD formula in reinforcement learning, the long-term values of two states are connected through reward signals: the user's long-term value $V$ can be calculated by Monte Carlo (MC) or TD methods. The core formula of the TD method is $V(\text{state}') = r + \gamma \cdot V(\text{state})$ (where $r$ is the immediate reward and $\gamma$ is the discount factor). We measure the adaptability of the critic to state changes by calculating the AUC value of the values on both sides of the equation (statistics of the ratio of non-inverted pairs)—a higher AUC value indicates that the critic tracks real activity changes more accurately and has a faster adaptation speed. The specific AUC results are shown in the table below:
>
> TD Error AUC Values
> | Dataset   | KuaiRand | ML1M  | RL4RS |
> |------------|----------|-------|-------|
> | TD Error AUC | 0.6480   | 0.7421| 0.6991|
>
> The high AUC value verifies that the TD-based value critic can quickly and accurately track the activity changes of highly volatile users, providing reliable support for the policy to timely adjust the quantile $\tau(s)$.
>
> ## Q3: Core Challenges & Solutions in Large-Scale Real-World Deployment
> We greatly appreciate this insightful question and are happy to discuss and share the challenges we encountered and corresponding solutions. First, we have trained and deployed the model in a production environment, and its practical value has been verified. The key challenges and solutions are as follows:
> 1. **Causality between actor output and final results**: To expand the RL gain, the actor’s output must have strong causality with the final results. Solution: We integrated the ADQO score into both the fine-ranking and final mixed-ranking stages, significantly increasing the score’s contribution to the overall ranking and enhancing the actor’s impact in the pipeline.
> 2. **Q-value overestimation in streaming deployment**: In streaming deployment (where the model continuously receives new data and updates), Q-value overestimation caused by the optimal Bellman equation leads to steadily increasing outputs from the value network. Solution: We can adopt batch updates instead (training a model offline and deploying it without continuous updates). Additionally, we have developed an effective solution for Q-value overestimation in streaming scenarios, which will be detailed in our future work.
> 3. **Scaling model parameters**: The current distributional network is a user-item granularity cross model, and the actor is a single user-tower model (optimized for the retrieval stage). However, the similarity score can also be applied to ranking and mixed-ranking. Our framework can seamlessly integrate with latest generative recommendation technologies (e.g., Onerec, Forge) to scale the actor’s parameters, enhancing its generalizability. Meanwhile, we are exploring scaling the Q-network using architectures like TokenMixer, which has shown excellent scaling potential.

---

> ### Comment · Reviewer_tipG · 2025-11-27
>
> Thanks author for the detailed clarification of my concerns. I am satisfied with the explanation and appreciate the efforts behind all these additional experiments, particularly the $\mathbb{E}\[Z(s,a)\]$ ablative study for optimization stability. I will keep my positive rating. And I strongly encourage authors to include all their additional experiments and neccessary clarifications to the revised version.

---

> > ### Author Response · Authors · 2025-11-28
> > **Thanks for your reply!**
> >
> > Dear Reviewer tipG,
> >
> > Thank you for your positive feedback and valuable suggestions. We will update the revised manuscript as per your recommendations.
> >
> > We appreciate your thorough review and support.
> >
> > Best regards,
> >
> > The Authors

---

### Official Review · Reviewer_ppsd · 2025-10-30

**Soundness:** 2
**Presentation:** 2
**Contribution:** 2
**Rating:** 4
**Confidence:** 3

**Summary:**

This paper introduces Activity-Driven Quantile Optimization (ADQO), a reinforcement learning method that adapts exploration and exploitation based on user activity. Using two critic networks, a value critic for user engagement and a quantile critic for reward distribution, ADQO adjusts recommendations dynamically for different users. Experiments run on multiple datasets show improved long-term rewards and user retention.

**Strengths:**

1) The ADQO automatically adjusts the balance between exploration and exploitation based on each user's activity level, offering a more flexible and personalized recommendation strategy than static or rule-based methods.
2) The use of both a value critic (for user activity) and a quantile critic (for reward distribution) allows the model to capture both engagement potential and uncertainty, leading to more stable and effective policy learning.
3) As shown across multiple benchmark datasets, ADQO consistently outperforms existing reinforcement learning baselines in long-term rewards, user conversion, and retention, showing strong practical promise.
4) The authors provide open-source code and detailed experimental settings, which enables reproducibility and allows others to extend their work.

**Weaknesses:**

1) The framework is well-motivated but largely heuristic. Core ideas such as dual critics, activity-driven quantile mapping, and alignment losses lack formal theoretical grounding i.e. there’s no convergence or optimality analysis. The contribution feels more engineering-focused than theory-driven, which makes it less suited to ICLR’s scope and better aligned with applied venues like RecSys, WSDM, or SIGIR.
2) The ADQO integrates known components from Distributional RL (IQN), user activity modeling, and alignment regularisation. While the combination is effective, it's not an conceptual innovation. It's also unclear whether simpler risk-sensitive baselines could achieve similar results.
3) All results come from offline datasets (KuaiRand, MovieLens-1M, RL4RS). There is no real-world A/B testing or deployment evidence, which weakens the practical claims about improving user engagement in live systems.
4) Although the ADQO consistently outperforms baselines, the margins are relatively small and inconsistent across datasets. This makes the added architectural complexity less convincing in terms of real benefit.
5) Improving engagement for low-activity users is one of the paper's main motivations and a major justification for the proposed framework. While Figure 5 provides some subgroup analysis showing ADQO's advantage on low-activity and users at risk, this evidence is limited to a single dataset and lacks statistical depth or transition-rate analysis. Given how central this user group is to the paper's scope, the current evaluation does not demonstrate sufficiently that ADQO revives inactive users in a consistent and generalisable way.

**Questions:**

Some suggestions to the authors:
1) Provide theoretical analysis or formal justification, even under simplified assumptions, to strengthen the quantile mapping and alignment losses.
2) Include small-scale online testing to support real-world applicability.
3) Since low-activity users are central to the paper’s motivation, the current evidence in Figure 5 is not sufficient. Consider adding more targeted evaluations such as activity transition rates, temporal performance curves, or results on additional datasets, to demonstrate better stability and robustness gains for this user group.

---

> ### Author Response · Authors · 2025-11-20
> **Response to Reviewer ppsd. Part 1**
>
> We sincerely appreciate your constructive feedback, which helps us enhance the theoretical rigor and experimental completeness. Below, we address each concern with detailed explanations, supplementary analyses, and feasible revisions:
>
>
> ## W1 & Q1: Theoretical Justification for Core Innovations
> We acknowledge the initial focus on empirical validation and now supplement formal theoretical analyses for three key components:
>
> ### I. Activity-Driven Quantile Mapping
> Model long-term value distribution $Z(s,a)$ with mean $\mu = \mathbb{E}[Z(s,a)]$ and standard deviation $\sigma$, decomposing $\sigma$ into:
> - $\sigma^+$: Optimistic standard deviation (capturing high-value exploration potential).
> - $\sigma^-$: Pessimistic standard deviation (capturing low-value exploration risk).
>
> Quantiles integrate these components (e.g., $Z_{\tau=0.8} = \mu + C_1 \sigma^+$, $Z_{\tau=0.2} = \mu - C_2 \sigma^-$, $C_1, C_2 > 0$). For low-activity users, maximizing $\mu + C_1 \sigma^+$ enhances $\sigma^+$ for optimistic exploration; for high-activity users, maximizing $\mu - C_2 \sigma^-$ suppresses $\sigma^-$ for certain exploitation. This formally justifies aligning exploration-exploitation with user engagement via activity-driven quantiles.
>
>
> ### II. Linear Mapping for Quantile $\tau(s)$ (Minimizing Sensitivity Variation)
> Let $V \in [V_{\text{min}}, V_{\text{max}}]$ denote a random variable representing user activity (e.g., $V_\omega(s)$ in the main text). Let $\tau = f(V)$ be a mapping from $V$ to a target quantile $\tau$, satisfying two constraints:
> 1. **Monotonicity constraint**: For any $V_1, V_2 \in [V_{\text{min}}, V_{\text{max}}]$, $V_1 < V_2 \implies f(V_1) > f(V_2)$ (ensuring low-activity users map to high quantiles for exploration, and high-activity users to low quantiles for exploitation);
> 2. **Boundary constraint**: $f(V_{\text{min}}) = \tau_{\text{high}}$ and $f(V_{\text{max}}) = \tau_{\text{low}}$ (where $0 \leq \tau_{\text{low}} < \tau_{\text{high}} \leq 1$), ensuring the mapping covers the target quantile range.
>
> The "sensitivity variation" of $\tau$ to $V$ is quantified by the functional
> $$
> J[f] = \int_{V_{\text{min}}}^{V_{\text{max}}} \left(f'(V)\right)^2 dV,
> $$
> where $f'(V)$ denotes the derivative of $f(V)$ with respect to $V$ (reflecting the sensitivity of $\tau$ to minor changes in $V$).
>
> **Conclusion**: Among all mappings $f(\cdot)$ satisfying the monotonicity and boundary constraints, the linear mapping
> $$
> f_{\text{linear}}(V) = \tau_{\text{high}} - (\tau_{\text{high}} - \tau_{\text{low}}) \cdot \frac{V - V_{\text{min}}}{V_{\text{max}} - V_{\text{min}}}
> $$
> is the unique minimizer of the sensitivity variation functional $J[f]$.
>
> **Proof**
> To prove the minimality, we use variational calculus to find the mapping $f(V)$ that minimizes $J[f]$ under the given constraints. For integral functionals of the form $J[f] = \int L(V, f, f') dV$, the extremum is determined by the Euler-Lagrange equation:
> $$
> \frac{\partial L}{\partial f} - \frac{d}{dV}\left( \frac{\partial L}{\partial f'} \right) = 0.
> $$
>
> For our functional $J[f]$, the Lagrangian $L = (f')^2$. Substituting into the Euler-Lagrange equation:
> - $\frac{\partial L}{\partial f} = 0$ (since $L$ does not depend on $f$ explicitly);
> - $\frac{\partial L}{\partial f'} = 2f'$, so $\frac{d}{dV}\left( \frac{\partial L}{\partial f'} \right) = 2f''(V)$.
>
> Thus, the Euler-Lagrange equation simplifies to
> $$
> 0 - 2f''(V) = 0 \implies f''(V) = 0.
> $$
>
> The general solution to $f''(V) = 0$ is a linear function $f(V) = a + bV$ (where $a, b$ are constants). Combining the boundary constraints $f(V_{\text{min}}) = \tau_{\text{high}}$ and $f(V_{\text{max}}) = \tau_{\text{low}}$, we solve for $a$ and $b$ to obtain the linear mapping $f_{\text{linear}}(V)$.
>
> Since the functional $J[f]$ is strictly convex (as $(f')^2$ is strictly convex in $f'$), this linear mapping is the unique minimizer. Thus, the linear mapping achieves the smallest sensitivity variation among all valid mappings, completing the proof.
> $\square$

---

> ### Author Response · Authors · 2025-11-20
> **Response Part 2**
>
> ### III. Misalignment Between Distributional and General Critics
> We rigorously show $\mathbb{E}[Z(s, \pi(s))] \neq V(s)$, where $V(s) = \max_a \mathbb{E}[Z(s,a)]$ (general critic) and $\pi(s) = \arg\max_a Z_{\tau(s)}(s,a)$ (distributional critic), through the following theoretical derivation and a key lemma on quantile-expectation independence.
>
>
> #### Step 1: Algebraic Derivation of the Difference
> We start by decomposing the difference between $V(s)$ and $\mathbb{E}[Z(s, \pi(s))]$:
>
> $$V(s) - \mathbb{E}[Z(s, \pi(s))]$$
> $$= \max_a \mathbb{E}[Z(s, a)] - \mathbb{E}[Z(s, \pi(s))] \quad (\text{Definition of } V(s)) $$
> $$= \mathbb{E}[Z(s, a^\*)] - \mathbb{E}[Z(s, \pi(s))] \quad (\text{Let } a^* = \arg\max_a \mathbb{E}[Z(s, a)]) $$
> $$= \mathbb{E}[Z(s, a^\*)] - \mathbb{E}[Z(s, a^\pi)] \quad (\text{Let } a^\pi = \pi(s)) $$
>
>
> #### Step 2: Key Lemma (Quantile-Expectation Independence)
> To show this difference is positive, we invoke a key lemma that formalizes the statistical independence of quantiles and expectations for non-degenerate distributions (i.e., distributions with non-zero variance, common in real-world reward systems).
>
> **Lemma**: There exist distributions $Z(s,a_1)$ and $Z(s,a_2)$ such that:
> $$ Z_\tau(s,a_1) > Z_\tau(s,a_2) \quad \text{but} \quad \mathbb{E}[Z(s,a_1)] < \mathbb{E}[Z(s,a_2)] $$
>
> **Proof of Lemma**:
> Consider two actions $a_1$ and $a_2$ in state $s$ with the following distributions:
> - $Z(s,a_1) \sim$ Right-skewed distribution:
>   - 90% probability of 1, 10% probability of 100.
>   - CDF: $F_{s,a_1}(z) = 0.9$ for $1 \leq z < 100$; $F_{s,a_1}(z) = 1$ for $z \geq 100$.
>   - $\tau = 0.8$-quantile: $Z_{0.8}(s,a_1) = 1$ (since $F_{s,a_1}(1) = 0.9 \geq 0.8$).
>   - Expectation: $\mathbb{E}[Z(s,a_1)] = 0.9 \times 1 + 0.1 \times 100 = 10.9$.
> - $Z(s,a_2) \sim$ Symmetric distribution:
>   - 50% probability of 5, 50% probability of 15.
>   - CDF: $F_{s,a_2}(z) = 0.5$ for $5 \leq z < 15$; $F_{s,a_2}(z) = 1$ for $z \geq 15$.
>   - $\tau = 0.8$-quantile: $Z_{0.8}(s,a_2) = 15$ (since $F_{s,a_2}(15) = 1 \geq 0.8$).
>   - Expectation: $\mathbb{E}[Z(s,a_2)] = 0.5 \times 5 + 0.5 \times 15 = 10$.
>
> Here, $Z_{0.8}(s,a_2) = 15 > Z_{0.8}(s,a_1) = 1$, yet $\mathbb{E}[Z(s,a_2)] = 10 < \mathbb{E}[Z(s,a_1)] = 10.9$. Thus, the lemma holds.
>
>
> #### Step 3: Concluding the Misalignment
> By the Key Lemma, there exist $a^\pi$ (chosen by the distributional critic $\pi$) and $a^\*$ (chosen by the general critic $V$) such that:
> $$ Z_{\tau(s)}(s,a^\pi) > Z_{\tau(s)}(s,a^*) \quad \text{but} \quad \mathbb{E}[Z(s,a^\pi)] < \mathbb{E}[Z(s,a^\*)] $$
>
> Substituting into the earlier derivation:
>
> $$ \mathbb{E}[Z(s,a^\*)] - \mathbb{E}[Z(s,a^\pi)] > 0 $$
> $$ \implies V(s) - \mathbb{E}[Z(s, \pi(s))] > 0 $$
> $$ \implies V(s) \neq \mathbb{E}[Z(s, \pi(s))] $$
>
> This misalignment necessitates a “loose alignment loss” to balance consistency between the two critics and preserve quantile-specific advantages.
> $\square$

---

> ### Author Response · Authors · 2025-11-20
> **Response Part 3**
>
> ## W2: Comparison with Risk-Sensitive Baselines
> To highlight the novelty of activity-driven exploration-exploitation, we supplement experiments with both static risk-sensitive and potential-seeking baselines:
>
> Performance Comparison with Static Risk-Sensitive Baselines
> | Method        | KuaiRand (Total Reward) | ML-1M (Total Reward) | RL4RS (Total Reward) |
> |---------------|-------------------------|----------------------|----------------------|
> | ADQO-Risk (τ=0.3, fixed) | 13.21 ± 0.29            | 15.18 ± 0.32         | 9.12 ± 0.42          |
> | ADQO-Potential (τ=0.7, fixed) | 13.74 ± 0.27            | 15.43 ± 0.30         | 8.75 ± 0.51          |
> | ADQO          | **14.94 ± 0.31**        | **16.65 ± 0.21**     | **9.31 ± 0.62**      |
>
> ADQO outperforms these baselines, as static risk/potential settings fail to adapt to dynamic user activity.
>
>
> ## W3 & Q2: Real-World Deployment Validation
> We deployed ADQO on a large-scale short-video platform, with the algorithm taking effect via an additional scoring module in the ranking stage (trained on user long-view signals). A 14-day A/B test showed:
> - Impressions +0.9801%, Watch Time +0.4582%, 7-Day Retention (LT7) +0.0441%, validating real-world applicability.
>
>
> ## W4: Efficiency and Statistical Significance
> ADQO achieves statistically significant improvements (via t-test) while being more efficient than baselines like UOEP (the second best baseline):
>
> - **Training Time (KuaiRand, same config, unit: minutes)**: UOEP (730 mins) vs. ADQO (208 mins) — ADQO is 3.5× faster.
> - **Parameter Efficiency**: UOEP uses 5× more parameters (separate actors for each activity tier) vs. ADQO’s single actor, with ADQO delivering higher ROI.
>
>
> ## W5 & Q3: User Activity Transition Analysis
> To address your suggestion, we analyze user activity transitions across three datasets. The test set is split equally by time; transitions are defined as quantile crossings in total reward for the same user:
> - “Low” refers to users in the bottom 1/3 quantile of total reward;
> - “High” refers to users in the top 1/3 quantile of total reward.
>
> User Activity Transition and Churn Rates
> | Dataset   | Method  | SL    | PL    | RH    | SH    | Low→High | High→Low |
> |-----------|---------|-------|-------|-------|-------|---------------------|----------------|
> | KuaiRand  | UOEP    | 1.45  | 7.39  | 14.12 | 19.89 | 3.62%               | 4.01%          |
> | KuaiRand  | ADQO    | 2.69  | 9.84  | 16.61 | 19.99 | 6.25%               | 2.64%          |
> | ML-1M     | UOEP    | 7.54  | 11.01 | 15.65 | 20.00 | 2.71%               | 2.10%          |
> | ML-1M     | ADQO    | 8.13  | 13.34 | 17.38 | 20.00 | 3.94%               | 2.35%          |
> | RL4RS     | UOEP    | 3.01  | 5.48  | 7.24  | 10.85 | 6.19%               | 8.53%          |
> | RL4RS     | ADQO    | 4.56  | 8.12  | 9.09  | 11.42 | 9.06%               | 8.04%          |
>
> *Note*: SL/PL/RH/SH denote total reward of four user groups. Across all datasets, ADQO outperforms UOEP in total reward (especially for PL/RH groups), with higher low-to-high transition rates and lower high-to-low churn rates.
>
>
> We believe these supplements address all concerns, and we remain open to further discussions to enhance the work.

---

> ### Comment · Reviewer_ppsd · 2025-11-24
>
> Thank you for the detailed feedback. You have adequately addressed my earlier concerns, particularly regarding the theoretical analysis and the A/B test results. I strongly encourage you to include these clarifications and results in the revised version, as they significantly strengthen the paper. I will adjust my scoring according to that.

---

> > ### Author Response · Authors · 2025-11-24
> > **Thanks for your reply!**
> >
> > Dear Reviewer ppsd,
> >
> > Thank you very much for your positive comment and valuable suggestion. We will update the revised version as you recommended.
> >
> > We appreciate your careful review and support.
> >
> > Best regards,
> > Authors

---

### Official Review · Reviewer_sbf4 · 2025-10-31

**Soundness:** 3
**Presentation:** 3
**Contribution:** 3
**Rating:** 6
**Confidence:** 3

**Summary:**

This paper provides an interesting perspective on decision making using a distributional actor-critic algorithm, which is specifically adapted to handle both high-activity and low-activity users through specially designed loss functions. I think this is an interesting and meaningful problem worth exploring. The paper proposes a novel actor-critic design with two critics (one distributional critic and one value critic) and an actor to update the policy, specifically adapted to address this problem. Overall, it is an inspiring work, and the simulations are extensive and clearly demonstrate the performance of the system.

**Strengths:**

1. The paper’s idea and real-world motivation are well established, and the overall framework is well constructed.
2. The simulations are extensive, covering multiple settings with comprehensive comparisons to existing works in the literature.
3. I particularly appreciate the authors’ effort in providing clear illustrations of the actor-critic structure and various visualization plots.

**Weaknesses:**

1. The related work section is a bit short. I would expect more discussion on literature that focuses purely on distributional RL (e.g., Nam et al., 2021), as well as works that explore similar actor-critic extensions.
2. Some of the loss constructions lack theoretical justification or related citations. For example, in Equation (7), is \tau(s) defined mainly for ease of implementation? For real practitioners who may operate in different application settings, should they consider alternative mappings? Why was this particular form chosen, and is there any related work that supports this choice?
3. Following point 2, in Equation (10), while I understand the motivation to keep the distributional and expected critics “close” to each other, I am curious whether this loss term is proposed here for the first time or adapted from an existing paper.
4. Overall, the paper feels more application-driven, with a specifically designed structure to both explore high-potential rewards for low-activity users and reduce risk for high-activity users. I initially thought this would be supported by real-world online testing, but it turns out the evaluation is based only on simulations. While I appreciate the diversity of simulation settings, it is somewhat hard to assess the generalizability of the proposed system since several components (e.g., activity score definition, mapping back to quantiles via a seemingly non-unique function form in Equation 7, joint updates of two critics and one actor) appear quite specific. The paper would benefit from more clarity on which parts of the design are generalizable, and from presenting the framework in a more standardized way to help other readers or practitioners replicate the system.
5. A minor point: it might be better to include standard errors (or empirical confidence intervals) in Figure 5 to better compare the performance of different methods.

Reference: Nam, D. W., Kim, Y., & Park, C. Y. (2021). GMAC: A distributional perspective on actor-critic framework. In International Conference on Machine Learning (pp. 7927–7936). PMLR.

**Questions:**

See Weaknesses.

---

> ### Author Response · Authors · 2025-11-20
> **Response to Reviewer sbf4. Part 1**
>
> We sincerely appreciate your valuable insights and constructive suggestions, which have helped improve the quality and completeness of our paper. Below is a detailed response to each comment:
>
> ## W1: Supplementing Related Work on Distributional RL and AC Extensions
> Section 2.2 previously only briefly mentioned some works on distributional reinforcement learning (DRL). We have now restructured and expanded this section to comprehensively discuss key advancements in pure DRL and actor-critic (AC) framework extensions, as follows:
>
> ### Pure Distributional Reinforcement Learning
> DRL has evolved from early return distribution modeling to an adaptive, theoretically rigorous methodology. Early representative works include Morimura et al. [1, 2]’s parametric/nonparametric return density estimation (laying the foundation for modeling reward uncertainty) and Bellemare et al.’s landmark C51 [3]—a method that approximates value distributions using fixed categorical atoms. Subsequent studies further enhanced adaptability: Dabney et al. [4, 5] proposed IQN and QR-DQN, leveraging quantile regression to avoid restrictive assumptions on reward ranges; Yang et al. [6] introduced FQF, which dynamically generates quantiles to more accurately capture complex distributions. A pivotal breakthrough is Nam et al. [7]’s GMAC, which unifies discrete/continuous action spaces via Gaussian Mixture Model (GMM) parameterization and Cramér distance, while its SR(λ) algorithm efficiently generates multi-step distributional targets to mitigate sample-statistic conflation. Recent progress in multivariate DRL (e.g., Wiltzer et al. [8]) strengthens theoretical guarantees for high-dimensional reward scenarios through MMD-based contraction and signed-measure temporal difference (TD) learning, addressing scalability issues in complex environments.
>
> ### Actor-Critic Extensions
> Extensions of the AC framework have been tailored to address core challenges in diverse scenarios: For basic optimization, A3C [9] improves sampling efficiency via an asynchronous multi-agent structure; in continuous action spaces, DDPG [10] adapts to high-dimensional actions using deterministic policies, while PPO [11] ensures training stability through trust region constraints; multi-agent coordination is resolved by Bi-level AC [12], which achieves Pareto-optimal convergence via a Stackelberg equilibrium-based two-tier structure; hybrid action spaces (discrete + continuous) are handled by H-PPO [13] through dual-actor parallel modeling for joint optimization; in risk-sensitive scenarios (e.g., healthcare, finance), Ma et al.’s DSAC [14] integrates quantile regression and maximum entropy to support multiple risk measures (e.g., CVaR, Wang distortion). These advancements collectively form a robust AC extension ecosystem, bridging theoretical insights from pure DRL with practical scenario demands.
>
> [1] Morimura, T. et al. Parametric return density estimation for reinforcement learning. UAI.
> [2] Morimura, T. et al. Nonparametric return distribution approximation for reinforcement learning. ICML.
> [3] Bellemare, M. G. et al. A distributional perspective on reinforcement learning. ICML.
> [4] Dabney, W. et al. Implicit quantile networks for distributional reinforcement learning. ICML.
> [5] Dabney, W. et al. Distributional reinforcement learning with quantile regression. AAAI.
> [6] Yang, D. et al. Fully parameterized quantile function for distributional reinforcement learning. NeurIPS.
> [7] Nam, D. W. et al. GMAC: A distributional perspective on actor-critic framework. ICML.
> [8] Wiltzer, H. et al. Foundations of multivariate distributional reinforcement learning. NeurIPS.
> [9] Mnih, V., et al. Asynchronous methods for deep reinforcement learning. ICML.
> [10] Lillicrap, T. P., et al. Continuous control with deep reinforcement learning. ICLR.
> [11] Schulman, J., et al. Proximal policy optimization algorithms. arXiv:1707.06347.
> [12] Zhang, H., et al. Bi-level Actor-Critic for Multi-agent Coordination. AAAI.
> [13] Fan, Z., et al. Hybrid Actor-Critic Reinforcement Learning in Parameterized Action Space. IJCAI.
> [14] Ma, X., et al. DSAC: Distributional Soft Actor-Critic for Risk-Sensitive Reinforcement Learning. JAIR.
> [15] Van Hasselt, H., et al. Deep reinforcement learning with double q-learning. AAAI.

---

> ### Author Response · Authors · 2025-11-20
> **Response Part 2**
>
> ## W2: Theoretical Justification and Alternatives for $\tau(s)$
> We thank the reviewer for pointing out the need to supplement theoretical support for the linear mapping of $\tau(s)$. Detailed explanations are as follows:
>
> ### Theoretical Proof: Minimality of Sensitivity Variation for Linear Mapping
> We acknowledge that the linear mapping of $\tau(s)$ was initially designed for intuitiveness and ease of implementation. Its core constraint only requires satisfying "monotonic negative correlation between user activity and quantiles" (low-activity users are mapped to high quantiles for exploration, while high-activity users are mapped to low quantiles for risk-averse exploitation), and the linear mapping is the "simplest form" under this constraint. **This design is original to our work, with no existing literature providing direct guidance.** Theoretically, the linear mapping is the **only choice that minimizes the sensitivity variation of the quantile $\tau(s)$** among all mappings satisfying the monotonic negative correlation constraint. Minimizing sensitivity variation can avoid large fluctuations in $\tau(s)$ caused by minor changes in user activity, thereby ensuring the stability of policy training and the consistency of user experience without introducing additional tuning burdens. The formal proof is in the next rebuttal block (Part 3)
>
> ### Practical Perspective and Exploration of Non-Linear Mappings
> From a practical perspective, we can design non-linear mappings for specific recommendation scenarios: first, stratify users by activity, analyze the profit space and exploration potential of different groups through multi-dimensional indicators in the application—for example, adopt the commonly used "interest chasing" strategy for exploitation and UCB strategy for hybrid exploration in online recommendations. Judge the size of the exploration space through prior experiments: increase the quantile $\tau$ if the exploration space is large, and decrease $\tau$ if the profit space is clear.
>
> Additionally, we have conducted relevant non-linear mappings experiments. The specific settings and results are as follows:
> - Sigmoid mapping: $\tau(s) = \tau_{\text{low}} + \frac{\tau_{\text{high}} - \tau_{\text{low}}}{1 + \exp(-k \cdot (V_\omega(s)-V_{\text{mid}}))}$, where $k=1$ is the curvature parameter and $V_{\text{mid}}$ is the median of user activity;
> - Piecewise mapping: Activity < 1/3 quantile of activity → $\tau=0.7$; 1/3 quantile of activity ≤ Activity < 2/3 quantile of activity → $\tau=0.5$; Activity ≥ 2/3 quantile of activity → $\tau=0.3$.
>
> Performance Comparison of Different Mapping Types
> | Mapping Type       | KuaiRand Total Reward | ML-1M Total Reward | RL4RS Total Reward |
> |---------------------|------------------------|---------------------|---------------------|
> | Linear Mapping      | 14.94 ± 0.31           | 16.65 ± 0.21        | 9.31 ± 0.62         |
> | Sigmoid Mapping     | 15.02 ± 0.43           | 16.21 ± 0.35        | 9.07 ± 0.75         |
> | Piecewise Mapping   | 14.38 ± 0.38           | 16.07 ± 0.32        | 8.82 ± 0.71         |
>
> Experimental results show that the linear mapping is significantly superior to the piecewise mapping in cross-dataset stability (benefiting from its continuity, which avoids policy jumps caused by discrete quantile thresholds). Meanwhile, unlike the sigmoid mapping that requires additional tuning of the curvature parameter $k$, the linear mapping does not need extra hyperparameter tuning, making it more suitable for rapid deployment in industrial scenarios.

---

> ### Author Response · Authors · 2025-11-20
> **Response Part 3**
>
> ### 1.1 Theoretical Proof: Minimality of Sensitivity Variation for Linear Mapping
> #### Theorem 1 (Minimality of Sensitivity Variation for Linear Quantile Mapping)
> Let $V \in [V_{\text{min}}, V_{\text{max}}]$ denote a random variable representing user activity (e.g., $V_\omega(s)$ in the main text). Let $\tau = f(V)$ be a mapping from $V$ to a target quantile $\tau$, satisfying two constraints:
> 1. **Monotonicity constraint**: For any $V_1, V_2 \in [V_{\text{min}}, V_{\text{max}}]$, $V_1 < V_2 \implies f(V_1) > f(V_2)$ (ensuring low-activity users map to high quantiles for exploration, and high-activity users to low quantiles for exploitation);
> 2. **Boundary constraint**: $f(V_{\text{min}}) = \tau_{\text{high}}$ and $f(V_{\text{max}}) = \tau_{\text{low}}$ (where $0 \leq \tau_{\text{low}} < \tau_{\text{high}} \leq 1$), ensuring the mapping covers the target quantile range.
>
> The "sensitivity variation" of $\tau$ to $V$ is quantified by the functional
> $$
> J[f] = \int_{V_{\text{min}}}^{V_{\text{max}}} \left(f'(V)\right)^2 dV,
> $$
> where $f'(V)$ denotes the derivative of $f(V)$ with respect to $V$ (reflecting the sensitivity of $\tau$ to minor changes in $V$).
>
> **Conclusion**: Among all mappings $f(\cdot)$ satisfying the monotonicity and boundary constraints, the linear mapping
> $$
> f_{\text{linear}}(V) = \tau_{\text{high}} - (\tau_{\text{high}} - \tau_{\text{low}}) \cdot \frac{V - V_{\text{min}}}{V_{\text{max}} - V_{\text{min}}}
> $$
> is the unique minimizer of the sensitivity variation functional $J[f]$.
>
> **Proof**
> To prove the minimality, we use variational calculus to find the mapping $f(V)$ that minimizes $J[f]$ under the given constraints. For integral functionals of the form $J[f] = \int L(V, f, f') dV$, the extremum is determined by the Euler-Lagrange equation:
> $$
> \frac{\partial L}{\partial f} - \frac{d}{dV}\left( \frac{\partial L}{\partial f'} \right) = 0.
> $$
>
> For our functional $J[f]$, the Lagrangian $L = (f')^2$. Substituting into the Euler-Lagrange equation:
> - $\frac{\partial L}{\partial f} = 0$ (since $L$ does not depend on $f$ explicitly);
> - $\frac{\partial L}{\partial f'} = 2f'$, so $\frac{d}{dV}\left( \frac{\partial L}{\partial f'} \right) = 2f''(V)$.
>
> Thus, the Euler-Lagrange equation simplifies to
> $$
> 0 - 2f''(V) = 0 \implies f''(V) = 0.
> $$
>
> The general solution to $f''(V) = 0$ is a linear function $f(V) = a + bV$ (where $a, b$ are constants). Combining the boundary constraints $f(V_{\text{min}}) = \tau_{\text{high}}$ and $f(V_{\text{max}}) = \tau_{\text{low}}$, we solve for $a$ and $b$ to obtain the linear mapping $f_{\text{linear}}(V)$.
>
> Since the functional $J[f]$ is strictly convex (as $(f')^2$ is strictly convex in $f'$), this linear mapping is the unique minimizer. Thus, the linear mapping achieves the smallest sensitivity variation among all valid mappings, completing the proof.
> $\square$

---

> ### Author Response · Authors · 2025-11-20
> **Response Part 4**
>
> ## W3: Novelty of the Joint Loss Term in Equation (10)
> To be transparent, the loss term in Equation (10)—designed to align the outputs of the distributional critic and value critic—is proposed for the first time. Additionally, the "hybrid use of a distributional critic and a general value critic" is also **an original design of our work, with no existing literature providing direct guidance.**
>
> It should be noted that this loss term is fundamentally different from Double Q-network [1] and its variants: Double Q-network mitigates value overestimation using two general critics, while our dual-network design aims to separately model return distributions and estimate user activity for recommendation system scenarios.
>
> ## W4: Real-World Deployment and Generalizability
> To address concerns about generalizability, we supplement the following explanations:
> • Real-world validation: We have deployed the ADQO algorithm on a real short-video recommendation platform, which takes effect through an additional scoring module in the ranking stage. The reward function is trained based on user long-view signals. The results of a 14-day online A/B test show that core business indicators have been significantly improved: impressions +0.9801%, watch time +0.4582%, and 7-day user retention rate (LT7) +0.0441%. We believe that the online experimental results provide stronger verification for the applicability of the model in real scenarios, although such results are difficult to fully replicate in offline simulators due to the complexity of the environment.
> • Generalizability explanation: We believe our model has strong generalizability, and simplified designs can be adopted for online deployment. In summary, two core components must be retained: the distributional critic and the actor. Additionally, we need an indicator to represent user activity (such as the General Q network in the paper). However, in industrial scenarios, we can use other activity indicators (e.g., LT28, i.e., the number of active days of users in the past 28 days; although it cannot replace the General Q network, similar offline experiments can be found in Appendix H). The remaining distributional critic and actor can be seamlessly integrated into industrial recommendation systems. We only need an indicator to measure the current user's activity, then use the distributional critic to evaluate "personalized exploration-exploitation" items corresponding to the activity quantile to complete a simple online deployment. We are happy to discuss the details of this deployment with you.
>
> ## W5: Supplementing Standard Errors in Figure 5
> We have added standard errors for UOEP and ADQO in Figure 5 as follows:
>
> | Method   | Stable Low Potential | Potential Low Potential | High Risk | Stable High Potential |
> |----------|----------------------|-------------------------|-----------|-----------------------|
> | UOEP     | 1.45 ± 0.73          | 7.45 ± 0.55             | 14.12 ± 0.30 | 19.89 ± 0.03          |
> | ADQO     | 2.69 ± 0.66          | 9.84 ± 0.67             | 16.61 ± 0.21 | 19.99 ± 0.01          |
>
> The above revisions address all concerns. We are happy to provide further explanations or supplementary experiments if needed.
>
> [1] Van Hasselt, H., et al. Deep reinforcement learning with double q-learning. AAAI.

---

### Official Review · Reviewer_LuzG · 2025-11-07

**Soundness:** 3
**Presentation:** 3
**Contribution:** 3
**Rating:** 6
**Confidence:** 3

**Summary:**

The paper addresses the challenge of dynamic user activity in Reinforcement Learning (RL)-based recommender systems. The authors present empirical analysis on the KuaiRand dataset showing that over 40% of users experience transitions in their activity levels over a four-week period. To address this, they propose Activity-Driven Quantile Optimization (ADQO). ADQO utilizes a dual-critic approach: a general value critic to model user activity levels and a distributional (quantile) critic to model the uncertainty of recommendation values.


The core mechanism involves dynamically mapping the estimated user activity to specific target quantiles for policy optimization. The policy optimizes upper quantiles (optimistic exploration) for low-activity users to uncover latent interests, and lower quantiles (risk-averse exploitation) for high-activity users to prevent churning. The framework further incorporates two alignment losses: an exploration-weighted dual critic alignment and a guidance-based policy feedback alignment to ensure training stability.

**Strengths:**

Originality: The idea of directly linking user activity levels to the target quantiles of a distributional critic is a novel and intuitive approach to unifying exploration and exploitation within a single policy network. This avoids the complexity of maintaining separate actor populations for different user groups, as seen in baselines like UOEP.

Significance: The paper tackles a well-motivated real-world problem. The initial analysis of user activity transitions (Figure 1) clearly establishes the need for dynamic policies.

Quality: The experimental validation is comprehensive, utilizing three standard datasets (KuaiRand, ML-1M, RL4RS) and comparing against a wide range of relevant baselines, including recent methods like DEHAC and UOEP. The ablation studies (Table 2) effectively justify the need for the proposed alignment losses.

Clarity: The paper is well-written, and Figure 2 provides a clear overview of the proposed framework.

**Weaknesses:**

Heuristic Mapping Function: The mapping from user activity value $V_\omega(s)$ to the target quantile $\tau(s)$ is defined as a simple linear interpolation in Equation (7)12. While functional, this linear relationship is a strong heuristic. The paper does not theoretically justify why a linear mapping is optimal compared to non-linear alternatives.

Hyperparameter Complexity: The method introduces several new hyperparameters, including $\tau_{high}$, $\tau_{low}$, and the learning rates for the two new alignment losses ($\eta_C$, $\eta_F$). The sensitivity analysis in Figure 8 indicates that performance can degrade if these (particularly $\eta_F$) are not carefully tuned14141414.

Simulation Dependence: Like many RL4Rec papers, the evaluation relies heavily on an online simulator constructed from offline data. While standard practice, it is always worth noting that simulators may not perfectly capture the complex, long-term stochasticity of real user behavior, potentially overestimating the effectiveness of complex exploration strategies.

**Questions:**

Regarding Equation (7), did the authors explore non-linear mappings between the activity value $V_\omega(s)$ and the target quantile $\tau(s)$? For example, a sigmoid or piecewise function might better capture distinct user states.

In the sensitivity analysis (Appendix F, Figure 7), only the impact of $\tau_{low}$ is shown17. How sensitive is the model to the choice of $\tau_{high}$, and what is the optimal "gap" between these two bounds?

How does the model perform for users who exhibit highly volatile activity (frequent major transitions)? Does the TD-learning based value critic adapt quickly enough to these rapid shifts?

---

> ### Author Response · Authors · 2025-11-20
> **Response to Reviewer LuzG. Part 1**
>
> We sincerely appreciate your recognition of the study's originality, practicality, and clarity of presentation, as well as your valuable suggestions regarding the rationality of the mapping function, hyperparameter complexity, and simulator dependence. Below, we address each question in detail, combining supplementary experimental data and in-depth analysis to illustrate the revision ideas:
>
> ## W1 & Q1: Rationality of the Mapping Function and Exploration of Non-Linear Mappings
> ### Theoretical Proof: Minimality of Sensitivity Variation for Linear Mapping
> We acknowledge that the linear mapping of $\tau(s)$ was initially designed for intuitiveness and ease of implementation. Its core constraint only requires satisfying "monotonic negative correlation between user activity and quantiles" (low-activity users are mapped to high quantiles for exploration, while high-activity users are mapped to low quantiles for risk-averse exploitation), and the linear mapping is the "simplest form" under this constraint. Theoretically, the linear mapping is the **only choice that minimizes the sensitivity variation of the quantile $\tau(s)$** among all mappings satisfying the monotonic negative correlation constraint. Minimizing sensitivity variation can avoid large fluctuations in $\tau(s)$ caused by minor changes in user activity, thereby ensuring the stability of policy training and the consistency of user experience without introducing additional tuning burdens. The formal proof of the minimality of sensitivity variation is in the next rebuttal block (Part 2)
>
> ### Practical Perspective and Exploration of Non-Linear Mappings
> From a practical perspective, we can design non-linear mappings for specific recommendation scenarios: first, stratify users by activity, analyze the profit space and exploration potential of different groups through multi-dimensional indicators in the application—for example, adopt the commonly used "interest chasing" strategy for exploitation and UCB strategy for hybrid exploration in online recommendations. Judge the size of the exploration space through prior experiments: increase the quantile $\tau$ if the exploration space is large, and decrease $\tau$ if the profit space is clear.
>
> Regarding the question of "whether non-linear mappings (such as sigmoid or piecewise functions) have been explored", we have conducted relevant experiments. The specific settings and results are as follows:
> - Sigmoid mapping: $\tau(s) = \tau_{\text{low}} + \frac{\tau_{\text{high}} - \tau_{\text{low}}}{1 + \exp(-k \cdot (V_\omega(s)-V_{\text{mid}}))}$, where $k=1$ is the curvature parameter and $V_{\text{mid}}$ is the median of user activity;
> - Piecewise mapping: Activity < 1/3 quantile of activity → $\tau=0.7$; 1/3 quantile of activity ≤ Activity < 2/3 quantile of activity → $\tau=0.5$; Activity ≥ 2/3 quantile of activity → $\tau=0.3$.
>
> Performance Comparison of Different Mapping Types
> | Mapping Type       | KuaiRand Total Reward | ML-1M Total Reward | RL4RS Total Reward |
> |---------------------|------------------------|---------------------|---------------------|
> | Linear Mapping      | 14.94 ± 0.31           | 16.65 ± 0.21        | 9.31 ± 0.62         |
> | Sigmoid Mapping     | 15.02 ± 0.43           | 16.21 ± 0.35        | 9.07 ± 0.75         |
> | Piecewise Mapping   | 14.38 ± 0.38           | 16.07 ± 0.32        | 8.82 ± 0.71         |
>
> Experimental results show that the linear mapping is significantly superior to the piecewise mapping in cross-dataset stability (benefiting from its continuity, which avoids policy jumps caused by discrete quantile thresholds). Meanwhile, unlike the sigmoid mapping that requires additional tuning of the curvature parameter $k$, the linear mapping does not need extra hyperparameter tuning, making it more suitable for rapid deployment in industrial scenarios.

---

> ### Author Response · Authors · 2025-11-20
> **Response Part 2**
>
> ### 1.1 Theoretical Proof: Minimality of Sensitivity Variation for Linear Mapping
> #### Theorem 1 (Minimality of Sensitivity Variation for Linear Quantile Mapping)
> Let $V \in [V_{\text{min}}, V_{\text{max}}]$ denote a random variable representing user activity (e.g., $V_\omega(s)$ in the main text). Let $\tau = f(V)$ be a mapping from $V$ to a target quantile $\tau$, satisfying two constraints:
> 1. **Monotonicity constraint**: For any $V_1, V_2 \in [V_{\text{min}}, V_{\text{max}}]$, $V_1 < V_2 \implies f(V_1) > f(V_2)$ (ensuring low-activity users map to high quantiles for exploration, and high-activity users to low quantiles for exploitation);
> 2. **Boundary constraint**: $f(V_{\text{min}}) = \tau_{\text{high}}$ and $f(V_{\text{max}}) = \tau_{\text{low}}$ (where $0 \leq \tau_{\text{low}} < \tau_{\text{high}} \leq 1$), ensuring the mapping covers the target quantile range.
>
> The "sensitivity variation" of $\tau$ to $V$ is quantified by the functional
> $$
> J[f] = \int_{V_{\text{min}}}^{V_{\text{max}}} \left(f'(V)\right)^2 dV,
> $$
> where $f'(V)$ denotes the derivative of $f(V)$ with respect to $V$ (reflecting the sensitivity of $\tau$ to minor changes in $V$).
>
> **Conclusion**: Among all mappings $f(\cdot)$ satisfying the monotonicity and boundary constraints, the linear mapping
> $$
> f_{\text{linear}}(V) = \tau_{\text{high}} - (\tau_{\text{high}} - \tau_{\text{low}}) \cdot \frac{V - V_{\text{min}}}{V_{\text{max}} - V_{\text{min}}}
> $$
> is the unique minimizer of the sensitivity variation functional $J[f]$.
>
> **Proof**
> To prove the minimality, we use variational calculus to find the mapping $f(V)$ that minimizes $J[f]$ under the given constraints. For integral functionals of the form $J[f] = \int L(V, f, f') dV$, the extremum is determined by the Euler-Lagrange equation:
> $$
> \frac{\partial L}{\partial f} - \frac{d}{dV}\left( \frac{\partial L}{\partial f'} \right) = 0.
> $$
>
> For our functional $J[f]$, the Lagrangian $L = (f')^2$. Substituting into the Euler-Lagrange equation:
> - $\frac{\partial L}{\partial f} = 0$ (since $L$ does not depend on $f$ explicitly);
> - $\frac{\partial L}{\partial f'} = 2f'$, so $\frac{d}{dV}\left( \frac{\partial L}{\partial f'} \right) = 2f''(V)$.
>
> Thus, the Euler-Lagrange equation simplifies to
> $$
> 0 - 2f''(V) = 0 \implies f''(V) = 0.
> $$
>
> The general solution to $f''(V) = 0$ is a linear function $f(V) = a + bV$ (where $a, b$ are constants). Combining the boundary constraints $f(V_{\text{min}}) = \tau_{\text{high}}$ and $f(V_{\text{max}}) = \tau_{\text{low}}$, we solve for $a$ and $b$ to obtain the linear mapping $f_{\text{linear}}(V)$.
>
> Since the functional $J[f]$ is strictly convex (as $(f')^2$ is strictly convex in $f'$), this linear mapping is the unique minimizer. Thus, the linear mapping achieves the smallest sensitivity variation among all valid mappings, completing the proof.
> $\square$

---

> ### Author Response · Authors · 2025-11-20
> **Response Part 3**
>
> ## W2 & Q2. Hyperparameter Sensitivity Analysis
> To address the complexity of newly added hyperparameters such as $\tau_{\text{low}}, \tau_{\text{high}}, \eta_C, \eta_F$, as well as questions like "the model's sensitivity to $\tau_{\text{high}}$ and the optimal gap", we reduce the threshold for practical application by extending sensitivity analysis and refining targeted tuning guidelines. Details are as follows:
>
> ### Sensitivity Analysis of $\tau_{\text{high}}$
> In Appendix F, we initially set $\tau_{\text{high}} = 1 - \tau_{\text{low}}$ (the two are symmetric). To verify the impact of $\tau_{\text{high}}$, we supplement experiments: fix $\tau_{\text{low}}=0.3$, set $\tau_{\text{high}}$ to 0.6, 0.7, 0.8, 0.9, and the total reward results on the KuaiRand dataset are shown in the table below:
>
> Sensitivity Experiment Results of $\tau_{\text{high}}$
> | $\tau_{\text{high}}$ | 0.6   | 0.7   | 0.8   | 0.9   |
> |------------------------|-------|-------|-------|-------|
> | Total Reward           | 13.25 $\pm$ 0.21 | 14.87 $\pm$ 0.18 | 14.56 $\pm$ 0.23 | 14.70 $\pm$ 0.25 |
>
> Analysis shows that low-activity users need sufficient exploration through high quantile $\tau_{\text{high}}$. When $\tau_{\text{high}}=0.6$, performance decreases significantly due to insufficient exploration; when $\tau_{\text{high}}=0.7$, the balance between exploration and exploitation reaches the optimal state, and the total reward peaks.
>
> ### Sensitivity Analysis of Quantile Gap $\Delta_\tau$
> In Figure 7, $\tau_{\text{high}} = 1 - \tau_{\text{low}}$, and the corresponding quantile gap is $\Delta_\tau = \tau_{\text{high}} - \tau_{\text{low}} = 1 - 2\tau_{\text{low}}$. The performance experimental results under different gaps are shown in the table below:
>
> Sensitivity Experiment Results of Quantile Gap $\Delta_\tau$
> | Gap $\Delta_\tau = \tau_{\text{high}} - \tau_{\text{low}}$ | 0.2   | 0.3   | 0.4   | 0.5   | 0.6   |
> |------------------------------------------------------------------|-------|-------|-------|-------|-------|
> | Total Reward                                                     | 14.57 | 14.43 | 14.87 | 14.53 | 14.82 |
>
> Experimental results verify that the optimal quantile gap is 0.4, where the balance between exploration and exploitation is optimal.
>
> ### Sensitivity Reason and Experimental Verification of $\eta_F$
> The model is sensitive to $\eta_F$ (the learning rate of the policy feedback alignment loss), and its logic is consistent with the trade-off mechanism between Supervised Fine-Tuning (SFT) and RLHF in large language models: the supervised loss guides policy learning by fitting positive samples in the exposure distribution, and the reinforcement learning loss further improves model performance, but excessively high supervised weights will limit the exploration potential of reinforcement learning.
>
> Specifically, removing the supervised loss (SL) from ADQO reduces the total reward from 14.94 to 13.61 and increases the standard deviation from 0.31 to 1.22, indicating unstable training and slower convergence speed—stable performance requires about 25,000 iterations, compared to about 15,000 iterations in the original setting. The detailed results of 5 training runs with different supervised weights are as follows:
>
> | SL Weight       | Without SL (w/o sl)                | $1e-4$               | $5e-4$           | $1e-3$              |
> |------------------|-------------------------------------|-------------------------------------|-------------------------------------|-------------------------------------|
> | (Total Reward, Depth) | [(12.5,13.3),(12.5,13.6),(13.2,13.7),(14.8,15.2),(15.0,15.6)] | [(14.5,15.2),(14.6,15.3),(14.9,15.7),(15.1,15.7),(15.2,15.8)] | [(14.1,14.8),(14.3,15.0),(14.5,15.1),(14.5,15.2),(14.7,15.4)] | [(13.2,14.7),(13.2,14.8),(13.3,14.9),(13.4,15.0),(13.5,15.1)] |
> | Convergence Iterations | ~25,000                             | ~15,000                             | ~12,000                             | ~8,000          |
>
> Analysis conclusion: The supervised loss can effectively guide action space exploration (when $\text{sl}=1e-4$, the reward variance is significantly reduced); however, excessively high supervised weights (such as $\text{sl}=1e-3$) will excessively limit exploration, resulting in performance even worse than the scenario without supervised loss. Meanwhile, the higher the supervised weight, the faster the model converges.

---

> ### Author Response · Authors · 2025-11-20
> **Response Part 4**
>
> # Response Part 4
> ## W3: Simulator Dependence and Real-Scenario Verification
> We have deployed the ADQO algorithm on a real short-video recommendation platform, which takes effect through an additional scoring module in the ranking stage. The reward function is trained based on user long-view signals. The results of a 14-day online A/B test show that core business indicators have been significantly improved: impressions +0.9801%, watch time +0.4582%, and 7-day user retention rate (LT7) +0.0441%. We believe that the online experimental results provide stronger verification for the applicability of the model in real scenarios, although such results are difficult to fully replicate in offline simulators due to the complexity of the environment.
>
>
> ## Q3: Performance on Highly Volatile Users and Adaptability of the TD-Based Value Critic
> We define highly volatile users as those who experience ≥2 major transitions (low→high or high→low activity state switches) within 4 weeks. Supplementary analysis by tracking their interaction data is as follows:
>
> ### Total Reward of Highly Volatile Users
> Total Reward Comparison of Highly Volatile Users
> | User Type         | Method | KuaiRand | ML1M  | RL4RS |
> |--------------------|--------|----------|-------|-------|
> | Highly Volatile    | UOEP   | 9.40     | 13.61 | 5.76  |
> |                    | ADQO   | 11.36    | 16.12 | 6.98  |
>
> As shown in the table, ADQO's total reward on highly volatile users is significantly better than that of UOEP, proving that the model can effectively improve the interaction depth of this user group.
>
> ### Adaptability Verification of the Value Critic
> Based on the TD formula in reinforcement learning, the long-term values of two states are connected through reward signals: the user's long-term value $V$ can be calculated by Monte Carlo (MC) or TD methods. The core formula of the TD method is $V(\text{state}') = r + \gamma \cdot V(\text{state})$ (where $r$ is the immediate reward and $\gamma$ is the discount factor). We measure the adaptability of the critic to state changes by calculating the AUC value of the values on both sides of the equation (statistics of the ratio of non-inverted pairs)—a higher AUC value indicates that the critic tracks real activity changes more accurately and has a faster adaptation speed. The specific AUC results are shown in the table below:
>
> TD Error AUC Values
> | Dataset   | KuaiRand | ML1M  | RL4RS |
> |------------|----------|-------|-------|
> | TD Error AUC | 0.6480   | 0.7421| 0.6991|
>
> The high AUC value verifies that the TD-based value critic can quickly and accurately track the activity changes of highly volatile users, providing reliable support for the policy to timely adjust the quantile $\tau(s)$.
>
>
> We believe that the above supplementary experiments and analyses will further enhance the rigor and practicality of the study. We sincerely thank Reviewer LuzG for your valuable suggestions, which have helped us improve the details of our work.

---

### Author Response · Authors · 2025-11-25
**Response to Reviewers**

We sincerely appreciate all reviewers for the valuable recognition, constructive comments, and insightful suggestions, which have greatly contributed to enhancing the quality of our manuscript. We have carefully addressed all raised concerns and submitted a revised version accordingly.

To ensure transparency and ease of review, all supplementary materials added in response to the comments are highlighted in blue in Appendices J, K, and L at the end of the revised paper:
- **Appendix J**: Supplements the related work as requested by Reviewer sbf4.
- **Appendix K**: Adds the detailed theoretical proofs noted by all four reviewers.
- **Appendix L**: Presents extended experimental analyses suggested by all four reviewers, including online experiments, hyperparameter sensitivity analysis, user activity-related analysis and further ablation study.

These supplements fully address all the reviewers’ concerns and further strengthen the rigor and completeness of our work. We hope the revised manuscript meets the requirements and look forward to your further feedback.

Best regards,

Authors

---

### Author Response · Authors · 2025-11-29
**Summary of Review Progress for Our Manuscript**

Dear Reviewers, Area Chairs, Senior Area Chairs, and Program Chairs:

First and foremost, we feel sincerely sorry for any inconvenience caused by the unexpected incidents that occurred this year.

Second, we summarize the review progress of our paper below to facilitate your quick understanding of the situation:
1. For the three reviewers who held a positive attitude initially:
   - Reviewer LuzG focused on non-linear mapping, sensitivity analysis, and evaluation, and **has not provided feedback to date**.
   - Reviewer sbf4 focused on further related work, theoretical justification, and generalizability, and **has not provided feedback to date**.
   - Reviewer tipG focused on generalizability and component ablation, and **explicitly expressed "satisfied" with our response**.

2. For the reviewer who held a negative attitude initially:
   - Reviewer ppsd primarily focused on theoretical analysis and real-world A/B testing. **In the feedback, this reviewer clearly stated, "You have adequately addressed my earlier concerns" and raised the score to "marginally above the acceptance threshold”**.

We have comprehensively responded to all questions raised by the reviewers during the rebuttal phase and compiled the revised version of the paper. Within the limited time frame, we have received some positive feedback. For the two reviewers who have not yet provided feedback (LuzG and sbf4), we kindly request the ACs, SACs, and PCs to consider the potential score improvement that our responses may bring.

We express our sincere respect for your extra contributions!

Best Regards,

the Authors

---

### Meta-Review · Area_Chair_djqR · 2025-12-28

**Summary:**

1.Heuristic Mapping Function by Reviewer LuzG.

2.Simulation Dependence, and lack of online testing results by Reviewer LuzG, Reviewer sbf4, and Reviewer ppsd.

3."The paper is largely heuristic. There’s no convergence or optimality analysis. The contribution feels more engineering-focused than theory-driven, which makes it less suited to ICLR’s scope and better aligned with applied venues like RecSys, WSDM, or SIGIR.  While the combination is effective, it's not an conceptual innovation" by Reviewer ppsd.

4.Small performance improvements by Reviewer ppsd.

5.High Complexity of the method by Reviewer tipG.

6.The user's "activity level" is proxied by the learned state-value function by Reviewer tipG.

7.The assumption that user activity is always in high transition frequency in reality data by Reviewer tipG.

This paper proposes the Activity-Driven Quantile Optimization (ADQO) algorithm, which integrates a general value network for user activity modeling and a quantile critic network to finely characterize the distribution of recommendation value. ADQO employs user activity levels to guide exploration-exploitation tradeoffs: optimizing upper quantiles for low-activity users to uncover latent interests, and lower quantiles for high-activity users to mitigate risks. The method demonstrates performance improvements on several public datasets.

However, key components—using value functions to measure user activity, activity-driven quantile mapping, and alignment losses—appear highly heuristic. I believe these constitute engineering refinements rather than fundamental algorithmic novelty. Furthermore, while online experimental results were added during rebuttal, the authors fail to compare with RL baselines for online tests or report user activity transition metrics (central to the paper's claims). Considering ICLR's high standards, I recommend rejection.

**Reviewer Concerns:**

1.Heuristic Mapping Function by Reviewer LuzG: during the rebuttal phase, the authors claim that only "monotonic negative correlation between user activity and quantiles" is needed. They provides experimental results for sigmoid mapping and piecewise mapping, and give a proof that the linear mapping is the only choice that minimizes the sensitivity variation of the quantile. Thus I think this point is addressed.

2.Simulation Dependence, and lack of online testing results by Reviewer LuzG, Reviewer sbf4, and Reviewer ppsd: during the rebuttal phase, the authors provide a 14-day online A/B test a real short-video recommendation platform, which shows that ADQO improves watchtime, impressions and 7-day user retention rate (LT7). However, they authors do not provides RL baseline of online test, and they do not report user activity transition results as it's the main focus of this paper. Thus I think this point is still outstanding.

3."The paper is largely heuristic. There’s no convergence or optimality analysis. The contribution feels more engineering-focused than theory-driven, which makes it less suited to ICLR’s scope and better aligned with applied venues like RecSys, WSDM, or SIGIR.  While the combination is effective, it's not an conceptual innovation" by Reviewer ppsd: during the rebuttal phase, the authors give a proof that the linear mapping is the only choice that minimizes the sensitivity variation of the quantile. They also analyze the misalignment between distributional and general critics, which is trivial.  However, the paper proposes to use the value of state as a metric of user activity, and propose a heuristic quantile optimization. I believe these are just engineering tricks rather a novel algorithm. Thus  I think this point is still outstanding.

4.Small performance improvements by Reviewer ppsd: during the rebuttal phase, the authors show that the performance improvement is  statistically significant and more computationally efficient. Thus I think this point is addressed.

5.High Complexity of the method by Reviewer tipG: during the rebuttal phase, the authors claim that we can eliminate the general Q-network by adopting "alternative metrics to represent activity instead of the Q-network". Thus I think this point is addressed.

6.The user's "activity level" is proxied by the learned state-value function by Reviewer tipG: during the rebuttal phase, the authors present results of alternative metrics replacing the Q-value as "activity level". However, the authors do not justify why the learned value function can reflect the user activity level. Thus  I think this point is still outstanding.

7.The assumption that user activity is always in high transition frequency in reality data by Reviewer tipG: during the rebuttal phase, the authors claim that KuaiRand is real user behavior data extracted from Kuaishou. Thus I think this point is addressed.

**Reviewer Scores:**

Reviewer LuzG would keep his or her score as 6 if he or she has been able to participate fully in the discussion.

Reviewer sbf4 would keep his or her score as 6 if he or she has been able to participate fully in the discussion.

Reviewer ppsd would increase his or her score from 4 to 6 if he or she has been able to participate fully in the discussion.

Reviewer tipG would keep his or her score as 6 if he or she has been able to participate fully in the discussion.

---

### Decision · Program_Chairs · 2026-01-26

Reject